

# A study of the influence of forest gaps on fire-atmosphere interactions

Michael T. Kiefer[1], Warren E. Heilman[2], Shiyuan Zhong[1], Joseph J. Charney[2], and Xindi Bian[2]

[1]Department of Geography, Michigan State University, East Lansing, MI 48824, USA
[2]USDA Forest Service, Northern Research Station, Lansing, MI 48910, USA

*Correspondence to:* Michael T. Kiefer (mtkiefer@msu.edu)

**Abstract.** Much uncertainty exists regarding the possible role that gaps in forest canopies play in modulating fire-atmosphere interactions in otherwise horizontally homogeneous forests. This study examines the impact of forest gaps on fire-atmosphere interactions using the ARPS-CANOPY model, a version of the Advanced Regional Prediction System (ARPS) model with a canopy pa-

rameterization. A series of numerical experiments are conducted with a stationary low-intensity fire, represented in the model as a line of enhanced surface sensible heat flux. Experiments are conducted with and without forest gaps, and with gaps in different positions relative to the fireline. For each of the four cases considered, an additional simulation is performed without the fire to facilitate comparison of the fire-perturbed atmosphere and the background state. Analyses of both mean and

instantaneous wind velocity, turbulent kinetic energy, air temperature, and turbulent mixing of heat are presented in order to examine the fire-perturbed atmosphere on multiple time scales. Results of the analyses indicate that the impact of the fire on the atmosphere is greatest in the case with the gap centered on the fire, and weakest in the case with the gap upstream of the fire. It is shown that gaps in forest canopies have the potential to play a substantial role in the vertical as well as horizontal

transport of heat away from the fire. Results also suggest that in order to understand how the fire will alter wind and turbulence in a heterogeneous forest, one needs to first understand how the forest heterogeneity itself influences the wind and turbulence fields without the fire.

## 1 Introduction

Wildland fires and the atmosphere interact across a range of spatial and temporal scales from macroscale

($10^5$ - $10^6$ m; hours to days) to microscale ($10^{-3}$ - $10^0$ m; seconds to minutes), and such interactions have been the subject of research for over a century [see Potter (2012a, b) for a review of the subject]. Studies of fire-atmosphere interactions have relevance for our understanding of (as well as modeling of) processes such as fire spread, smoke transport/dispersion, and tree mortality. Previous studies of fire-atmosphere interactions have mainly focused on horizontally homogeneous forests; the im-

pact on fire-atmosphere interactions of gaps or openings in otherwise homogeneous forests, whether



natural (e.g., windstorm damage) or man-made (e.g. fuel breaks), remains largely unexplored. Furthermore, it is unclear how fire-induced atmospheric perturbations evolve as the fire progresses from forest to gap and back to forest. Note that the term "gap" is used in this study to denote clearings or overstory fuel breaks with horizontal dimensions of approximately one canopy height or larger, and
not spaces between branches or between individual trees.

Before proceeding to discussion of the current state of knowledge of fire-atmosphere interactions inside forest gaps, some discussion of the simpler case of fire-atmosphere interactions in homogeneous canopies is warranted. The impact of homogeneous forest canopies on simulated fire-perturbed variables [e.g., temperature, turbulent kinetic energy (TKE)] and processes affecting such
variables [e.g., turbulent mixing, shear production] was examined in Kiefer et al. (2015). In a series of numerical experiments within a homogeneous forest, the sensitivity of mean and turbulent flow downstream of a low-intensity surface fire to canopy density was examined. In general, near-surface turbulence both prior to and during the fire was reduced in the presence of a canopy. Both the fireline normal component of wind and maximum vertical velocity were shown to be weaker with a
sparse canopy than with no vegetation, although both variables were largely insensitive to further increases in canopy density. However, the influence of the fire on planetary boundary layer (PBL)-integrated vertical turbulent heat flux was greatest with a sparse canopy and gradually weakened with increasing canopy density.

With regard to fire-atmosphere interactions within gaps, although some modeling studies have
examined fire propagation in discontinuous fuels beds (e.g., Linn et al., 2005), only Pimont et al. (2009, 2011) have examined in detail simulated fire-atmosphere interactions inside forest gaps. Pimont et al. (2009) examined the impact of a 180-m wide fuel break on mean and turbulent flow using the FIRETEC model (Linn and Cunningham, 2005) and simulated higher mean wind velocity and gust intensity inside the fuel break than in the surrounding forest, but reduced variability of wind
direction in the break, relative to the surrounding forest. Pimont et al. (2011) used the FIRETEC model to examine the sensitivity of wind flow and fire propagation to structural parameters of a multiple grid-level forest canopy, including tree-cover fraction and clump size. Varying the structural parameters within a 200-m wide "treated zone" or fuel break and igniting a fire line upwind of the treated zone, the authors showed wind velocities to be stronger in the treated zone, relative to the
surrounding forest, and found that the plume of hot gases was most strongly tilted from vertical when the fireline was inside the treated zone.

Studies that focus on flow within forest gaps in the absence of fire are somewhat greater in number (e.g., Bergen, 1975, 1976; Schlegel et al., 2012, 2015; Queck et al., 2015), though still small relative to the number of studies of flow within homogeneous forests and orchards (e.g., Shaw and Schu-
mann, 1992; Raupach et al., 1996; Watanabe, 2004; Dupont and Brunet, 2008; Dupont and Patton, 2012a, b; Kiefer et al., 2015) and near forest edges (e.g., Patton et al., 1998; Lee, 2000; Dupont and Brunet, 2007, 2009). Studies of flow within forest clearings or gaps consist primarily of field ex-





periments (e.g., Bergen, 1975, 1976; Queck et al., 2015) and large-eddy simulation (LES) modeling (e.g., Schlegel et al., 2012, 2015; Queck et al., 2015). Analysis of field experiment data and LES

modeling results in Queck et al. (2015) showed the impact of small inhomogeneities such as forest clearings on not only turbulent flow but also mean flow within the canopy. A recirculation zone was simulated with the LES model, consistent with other LES studies of flow in forest clearings (e.g., Schlegel et al., 2012, 2015), which was confirmed by companion field experiments, but was not reproduced in companion wind tunnel experiments and Reynolds Averaged Navier Stokes (RANS)

modeling. The recirculation zone was found to be largely confined to the clearing, with the influence of the clearing on mean flow quickly diminishing with height above the canopy.

In this study, we use the ARPS-CANOPY model (Kiefer et al., 2013) to explore the sensitivity of fire-perturbed atmospheric variables (e.g., air temperature, turbulent kinetic energy) to the presence of gaps in the forest cover. ARPS-CANOPY is a modified version of the Advanced Regional

Prediction System (ARPS) model (Xue et al., 2000, 2001) in which the effects of vegetation elements (e.g., branches, leaves) on drag, turbulence production/dissipation, radiation transfer, and the surface energy budget are accounted for through modifications to the ARPS model equations. ARPS-CANOPY has been applied to simulations of real-world prescribed fires in heterogeneous forest canopies (Kiefer et al., 2014) and idealized low-intensity fires in homogeneous forest canopies

(Kiefer et al., 2015). Note that due to limitations of ARPS-CANOPY, this study does not directly address the impact of forest gaps on processes such as fire spread and tree injury. However, ARPS-CANOPY is suited to the specific goal of this study: examining how gaps in forest canopies impact fire-atmosphere interactions. Lastly, note that unlike Pimont et al. (2009) and Pimont et al. (2011), our focus is on fire-atmosphere interactions in general, and not the specific impacts of gaps on fire

behavior. Furthermore, this study considers not only how fire-atmosphere interaction in forest gaps differs from the more studied homogeneous forest case, but also whether the ability of the fire to perturb the atmosphere is sensitive to the position of the gaps relative to the fire (e.g., upstream vs. downstream gap).

The remainder of this paper is organized as follows. A description of the model and experiment

design are included in Section 2, including a brief overview of the ARPS-CANOPY model and how it differs from the standard ARPS model (2.1), a description of the model configuration and parameterization (2.2), and a summary of the experiment design (2.3). Results and discussion of the sensitivity experiments are presented in Section 3, beginning with a brief summary of the analysis methodology (3.1), followed by analysis of mean (3.2) and instantaneous (3.3) variables. Finally, the

paper is concluded in Section 4.





## 2 Model description and numerical experiment design

### 2.1 ARPS-CANOPY Overview

The development of ARPS-CANOPY is described in detail in Kiefer et al. (2013); the following is
a brief summary. For validation of ARPS-CANOPY in orchard and forest environments, see Kiefer
et al. (2013) and Kiefer et al. (2014), respectively.

First, modifications to the ARPS model equations were made to account for the drag force of
vegetation elements, via a drag force term added to the momentum equation, and the enhancement
of turbulence dissipation in the canopy air space, via a sink term added to the sub-grid scale (SGS)
turbulent kinetic energy (TKE) equation (Dupont and Brunet, 2008). Note that although multiple
turbulence schemes are available in ARPS, ARPS-CANOPY exclusively utilizes the 1.5 order TKE-
based turbulence closure (Deardorff, 1980; Moeng, 1984).

Subsequently, Kiefer et al. (2013) modified ARPS-CANOPY to allow for simulation of non-
neutral canopy flows. Specifically, a term was added to the thermodynamic equation to represent
heating (cooling) of the canopy air spaces resulting from the vertical flux convergence (divergence)
of net radiation intercepted by the canopy, and the ground radiation budget was modified to account
for shading of the ground surface by the overlying vegetation during the day and reduction of out-
going longwave ground radiation at night. As an alternative to computing the radiation budget at
all grid points within the canopy, a net radiation profile was employed that decays downward from
canopy top as a function of the cumulative leaf area index (computed from the top of the canopy
downward) and an empirically determined extinction coefficient [0.6 in this study, as in e.g., Kiefer
et al. (2013); Dupont and Brunet (2008)]. Computation of the net radiation budget at canopy top is
otherwise identical to the standard ARPS ground radiation budget, except a constant value of albedo
appropriate for forested areas is utilized (0.1), and the outgoing longwave component is computed
as a function of air temperature at canopy top, rather than skin temperature. Lastly, a production
term was added to the SGS TKE equation to represent turbulence production in the wakes of canopy
elements.

It is important to note that ARPS-CANOPY does not resolve the flow around individual trees
or the heating/cooling of individual branches or leaves. In all aspects of the model, the canopy is
represented as a height-varying plant area density profile ($A_p$), specified at each grid point. $A_p$,
defined as the one-sided area of all plant material per unit volume of canopy, is a bulk measure of
the density of a large group of trees. It is also important to note that aside from the modifications
outlined here, ARPS-CANOPY is otherwise identical to standard ARPS.

### 2.2 Model configuration and parameterization

As stated in the previous section, a 1.5-order subgrid-scale turbulence closure scheme with a prog-
nostic equation for TKE is utilized. Radiation physics following Chou (1990, 1992) and Chou and



Suarez (1994) are applied outside of the canopy, with the parameterization outlined in Section 2.1 applied at points inside the canopy. Moist processes are represented in the model, with Lin ice microphysics (Lin et al., 1983) and explicit convection enabled. Fourth-order accurate finite differencing of the advection terms is used in both the vertical and horizontal directions. The Coriolis force is
computed (as a function of central latitude only).

A 1-way nesting procedure is utilized with two three-dimensional computational domains, and a periodic boundary condition is applied at the lateral boundaries of the outer domain. The outer domain consists of 153 x 103 x 78 grid points (including points used only for boundary condition calculations), with 50 m horizontal grid spacing; the inner domain, centered within the outer domain,
consists of 99 x 51 x 78 grid points, with 10 m horizontal grid spacing. Vertical grid spacing of 2 m is utilized in both domains, up to a height of 84 m, above which vertical stretching is applied. With this vertical grid structure, there are 9 grid points at or below the canopy crown (canopy height is 18 m). The top of both model domains is at 3 km, with a rigid lid upper boundary condition and a Rayleigh damping layer in the uppermost 1 km to prevent reflection of waves from the upper boundary.

The outer domain simulation is initialized at noon local time, with a uniform net radiation flux of 520 $\mathrm{Wm}^{-2}$ applied at the canopy top to represent daytime heating typical of $40^o$ N latitude in early spring. The outer domain simulation is run with a uniform canopy (see Section 2.3) and no fire, and is initialized with a base state sounding consisting of uniform wind speed (2.5 $\mathrm{ms}^{-1}$, westerly) from the surface to domain top and neutral static stability below z = 1 km (stable stratification
above). Although the model is initialized with a horizontally homogeneous atmosphere, a random perturbation of magnitude 1 K is applied to the potential temperature field at the initial time (at all model levels) to promote the development of 3D turbulent structures. The outer domain simulation is run for a total of four hours; after approximately three hours a horizontally quasi-homogeneous and quasi-stationary PBL develops, and the inner domain simulation is initialized at that end of hour
three. The inner domain simulation is run for one hour, with boundary conditions updated every five minutes.

## 2.3   Experiment design

Following a 30-minute spin-up period, a 25 $\mathrm{kWm}^{-2}$ surface turbulent sensible heat flux, representative of a low-intensity fire, is applied within a 50-m wide north-south strip positioned 3.2 km
downstream of the western boundary of the inner domain. The heat from the fire is laterally distributed across the strip in a step pattern, with 85% of the total heat flux (21.25 $\mathrm{kWm}^{-2}$) applied at the center grid point, and the fire-induced heat flux in the flanking cells stepping down to zero three cells away. Recall from the previous section that a 2.5 $\mathrm{ms}^{-1}$ westerly base state wind is applied in all simulations; therefore, the ambient wind is perpendicular to the fireline.

The portion of the domain surrounding the fire line is divided into three zones, delineated by their position relative to the fire center: **U**pstream, **C**enter, and **D**ownstream (Fig. 1). Four canopy





configurations are considered in this study: uniform canopy with no gaps (case NG), gap in the upstream zone (UG), gap in the center zone (CG), and gap in the downstream zone (DG); for all gap experiments the gap is 50 m (5 grid points) wide. A summary of the canopy gap configurations is provided in Fig. 1a, along with the $A_p$ profile used in all experiments, and a vertical cross-section of $A_p$ in the x-z plane is provided in Fig. 1b (case CG only). Note that within the forest blocks the canopy is horizontally homogeneous and represented by a single $A_p$ profile, and within the gaps $A_p$ is zero; the resulting $A_p$ cross-section (Fig. 1b) is similar to case HOM in Schlegel et al. (2012). The vertical canopy profile is characterized by a moderately dense overstory (max $A_p = 0.455 \ \mathrm{m}^{-2}\mathrm{m}^{-3}$), and an open trunk space with sparse understory (min $A_p = 0.006 \ \mathrm{m}^{-2}\mathrm{m}^{-3}$); the plant area index (PAI) is 2 and the canopy height (h) is 18 m. Such a profile is typical of, for example, Maritime pine (Pinus pinaster) and Loblolly pine (Pinus taeda) stands. Finally, note that two simulations are performed for each canopy configuration, one with a parameterized fire and one without (referred to hereafter as "fire" and "no-fire" simulations).

## 3 Results and discussion

### 3.1 Analysis methodology

Perturbation wind components ($u'$, $v'$, $w'$) and temperature ($T'$) are computed by subtracting 30-min mean quantities from the no-fire simulation from the instantaneous values. Use of the no-fire mean ensures that modification of the mean variables by the fire does not influence the calculation of perturbations. Instantaneous TKE is subsequently computed as $\frac{1}{2}\left(u'^2 + v'^2 + w'^2\right)$ and turbulent heat fluxes are computed as $u'T'$ (horizontal) and $w'T'$ (vertical). In this study, spatiotemporal mean quantities in a layer of depth 3*h are examined first (Section 3.2), followed by analysis of instantaneous variables inside the canopy layer (Section 3.3). For the mean variable analysis (Section 3.2), turbulent and mean quantities are averaged along the y-axis and over the 30-min period during which the fire is applied. Subsequently, vertical (x-z) cross sections of the difference between the fire and no-fire mean quantity (hereafter, the fire anomaly) are displayed along with the corresponding no-fire mean quantity. For the instantaneous variable analysis (Section 3.3), box and whisker plots of instantaneous variables within the canopy layer are compared between the three zones (**U**pstream, **C**enter, and **D**ownstream) for each case (NG, CG, UG, and DG).

### 3.2 Mean variable analysis

Examination of the four cases begins with vertical cross-sections of the spatiotemporal mean u-component of the wind (Fig. 2); see Section 2.3 for a description of the averaging procedure. Consider the no-fire wind field in the absence of fire first (contour lines). In the no gap case (Fig. 2a), one finds a weak wind regime within the canopy and a layer of pronounced vertical wind shear centered near the top of the canopy (green line). Introduction of the gap (Figs. 2b-d) yields a near-surface





wind reversal within the gap (i.e., recirculation zone) and a tongue of stronger winds penetrating from above the canopy into the gap. Such a wind flow pattern is consistent with flux tower observations and LES modeling of flow through a 60 m wide gap as reported in Schlegel et al. (2012) and Queck et al. (2015). With the fire implemented (color shading), a broadly similar pattern in the fire

anomaly field is seen in all four cases: positive anomalies centered on the fire, and negative anomalies mainly 100-200 m downstream of the fire center. The magnitude of the anomaly is sensitive to the position of the gap relative to the fire: the largest fire anomaly occurs in the case where the the fire zone is co-located with the recirculation zone (case CG) and the weakest fire anomaly occurs in the case where the fire zone is co-located with the transition zone downstream of the gap where the

westerly flow through the canopy in the absence of fire is stronger than corresponding flow through a homogeneous canopy (case UG).

Proceeding to vertical cross-sections of vertical wind velocity (Fig. 3), a weak and nebulous field of vertical motion is evident above the uniform canopy (case NG) in the absence of fire (Figs. 3a). In the continued absence of any fire, introduction of the gap (Figs. 3b-d) yields a vertical motion

field (contour lines) consisting of a compact updraft on the upwind (i.e., west) side of the gap (solid lines) and a broad downdraft on the downwind (i.e., east) side of the gap (dashed lines). The vertical velocity couplet completes a clockwise circulation within the gap (cf. Figs. 2,3) that has been documented in previous studies (Schlegel et al., 2012; Queck et al., 2015). Small differences in the magnitude of the circulation between gap cases is likely evidence of the impact of transient distur-

bances in the unstable boundary layer on flow within the gap (not shown). Introduction of the fire yields a fire anomaly couplet of broader scale than the vertical velocity couplet associated with the gap itself, consisting of a negative (positive) fire anomaly on the upwind (downwind) side of the fire. Notably, the positive fire anomaly appears to be less sensitive to the position of the gap relative to the fire than the negative anomaly, which is weakest in case UG and strongest in case CG. The

strongest negative anomaly occurs in the case wherein the upwind side of the fire (and fire-induced downdraft) coincides with the upwind side of the gap (and associated updraft), i.e., in case CG.

Examining total (resolved plus SGS) TKE in the absence of fire or gap (Fig. 4a, contours), a three-layer structure is seen, with an above-canopy maximum between 30 and 40 m AGL, a secondary in-canopy maximum at the surface, and a minimum in the upper canopy. Introduction of the gap induces

a largely homogeneous TKE field within the gap and a TKE maximum above the gap, with a tongue of higher TKE penetrating into the clearing from above; the influence of the gap on the TKE spatial structure diminishes rapidly with downstream distance from the clearing. As with the horizontal and vertical velocity fields, the TKE pattern inside and out of the gap is consistent with the LES study of Schlegel et al. (2012) (their Fig. 10a; case HOM). With the fire heat source engaged (color shading),

a fire anomaly maximum develops over or immediately downstream of the fire center. The largest fire anomaly occurs in case CG and the weakest anomaly occurs in case UG, mirroring the fire anomaly



pattern for the u-component of the wind (Fig. 2). Analysis of the resolved TKE budget (not shown) indicates that buoyancy production is the primary source of the near-surface TKE anomaly.

Proceeding to temperature in the absence of fire or gap (Fig. 5a), a superadiabatic lapse rate is
evident ($\sim 3.75$ K 100 m$^{-1}$), along with a weak horizontal gradient related to transient features in the unstable boundary layer (not shown). With the gap introduced, a pronounced horizontal gradient develops across the gap, with the axis of coolest temperatures just downstream of the gap. The influence of the gap on the magnitude of the fire anomaly is more modest than with the other variables, with only about a 4% difference in anomaly magnitude between the cases with the largest
(CG) and smallest (UG) anomalies. However, the downstream limit of the anomaly zone is sensitive to gap position, with anomalies extending farthest downstream in case UG. Differences in the downstream extension of temperature anomalies are attributable to differences in the magnitude of temperature advection downstream of the fire, a claim that is supported by a close examination of the u-component of the wind in each case (Fig 2b).

As this is a study of fire-atmosphere interactions in heterogeneous forests, a relevant question to ask is what role, if any, do gaps play in vertical heat transport? In other words, do gaps act as vents for hot gases? To help answer this question, we examine vertical (horizontal) turbulent mixing as represented by the vertical (horizontal) gradient of vertical (horizontal) turbulent heat flux, in a vertical cross section (Figs. 6 and 7). The pattern of vertical and horizontal turbulent mixing anomaly
is the same in all cases: in the vertical, heat is mixed upward away from the surface heat source, and in the horizontal, heat is mixed in the downstream direction. As was the case for the other variables examined (e.g., u-component of the wind, vertical velocity), the strongest and weakest anomalies occur in cases CG and UG, respectively. In other words, the strongest turbulent mixing occurs when the gap is positioned above the fireline, and the weakest turbulent mixing occurs when the gap
is positioned upstream of the fireline. These results indicate that gaps in forest canopies have the potential to play a substantial role in the vertical as well as horizontal transport of heat away from the fire.

### 3.3 Instantaneous variable analysis

Acknowledging that fire-atmosphere interactions occur on a variety of temporal and spatial scales,
we proceed from analysis of time and space-averaged quantities to analysis of instantaneous variables inside the canopy in the presence of fire. The analysis begins with the instantaneous u-component of the wind (Fig. 8) in case NG (Fig. 8a). In zone U, upstream of the fire, the u-component inside the canopy is in the range -1.88 to 2.67 ms$^{-1}$, with a median value of $0.48$ ms$^{-1}$. In zones C and D, over or downstream of the fire, the range of wind speeds is considerably larger [C: u $\in$ (-3.08,3.74); D: u
$\in$ (-3.09,3.44)] and the median is about 40% larger than in zone U. Thus, it appears that the fire has a pronounced effect on the magnitude of horizontal wind gusts as well as the intensity of the mean wind within the canopy. Examining cases UG, CG, and DG (Fig. 8b-d), it is evident that removal of



vegetation from a zone increases both the width of the distribution and the median, especially when the gap is over or downstream of the fire (e.g., zone C in case CG, zone D in case DG). Wind speeds

are larger wherever a gap is present due to reduced drag and turbulent dissipation.

We next proceed to instantaneous vertical velocity (Fig. 9). A quick examination of all cases and zones shows that magnitudes of instantaneous vertical velocity inside the canopy are generally between 1.5 and 3.75 $\mathrm{ms}^{-1}$, about an order of magnitude larger than the magnitude of mean vertical velocity seen in Fig. 3. It is worth noting, however, that even larger instantaneous vertical velocities

are present above the canopy within the mixed layer [O(5-10 $\mathrm{ms}^{-1}$)] (not shown). Proceeding to case NG (Fig. 9), we find that unlike the quasi-normal distribution of the u-component of the wind, there is a positively skewed distribution of vertical velocity in zone U. Over and downstream of the fire (zones C and D), the width of the distribution increases as well as the skewness. As with the u-component of the wind, introduction of a gap in a particular zone increases the width of the

distribution, especially when the gap is over or downstream of the fire. However, the effect of the gap on the median vertical velocity value is ambiguous, as the median in some cases remains unchanged (e.g., compare zone C in cases NG and CG; Figs. 9a,c) and in others switches sign (e.g., compare zone D in cases NG and DG; Figs. 9a,d).

Examination of instantaneous TKE (Fig. 10) in case NG (Fig. 10a) reveals a median value of

0.25 $\mathrm{m^2s^{-2}}$ in zone U, but outlier values as large as 6.92 $\mathrm{m^2s^{-2}}$. Over and downstream of the fire (zones C and D), the median is more than double the zone U value, and outlier values increase to as large as 8.81 $\mathrm{m^2s^{-2}}$ in zone C. The introduction of forest gaps yields a considerably more turbulent atmosphere, as judged by median and outlier values. When the gap is centered on the fire, a median TKE of 0.91 $\mathrm{m^2s^{-2}}$ is produced, with instantaneous values as large as 14.93 $\mathrm{m^2s^{-2}}$. Similar

conditions occur when the gap is positioned downstream of the fire (case DG); note that the largest median value in any zone occurs in zone D in case DG (1.08 $\mathrm{m^2s^{-2}}$). It is clear from this analysis that canopy gaps serve to locally increase turbulence in an otherwise drag-tempered environment. Furthermore, large TKE values may occur inside a gap in the forest canopy even though the gap is not directly above the fire.

Lastly, we examine instantaneous temperature (Fig. 11). Beginning with case NG (Fig. 11), the influence of the fire on temperature is evidenced by the nearly 27 K difference in maximum temperature between zones U (307.94 K) and C (334.79 K), as well as the 1.3 K difference in median temperature values between the two zones. Whereas this effect of the fire on temperature is expected, the influence of the gap on temperature is, however, less intuitive. In contrast to the other variables

examined (e.g., TKE), the instantaneous temperatures in a particular zone (as judged by the maximum and median values) either remain unchanged or actually decrease when a gap is introduced. The result that near-surface temperatures above a fire are less hot in the absence of vegetation, as compared to a homogeneous forest, is consistent with the findings of Kiefer et al. (2015). Recall that the same amount of heat is fluxed into the atmosphere per second per unit area (25 $\mathrm{kWm}^{-2}$);



introduction of the gap increases the intensity of turbulent mixing and serves to reduce the maximum
temperature compared to uniform vegetation.

## 4 Summary and conclusions

In this study, ARPS-CANOPY has been utilized to examine the sensitivity of fire-perturbed variables
to the presence of gaps in forest cover, and to the position of such gaps relative to the fire line. A

single plant area density profile was used to represent a canopy with a moderately dense overstory
and sparse trunk space, and a 25 $\mathrm{kWm}^{-2}$ heat flux was applied within a 50-m wide fireline to
represent a low-intensity fire. Simulations with and without the fire were performed to evaluate the
role of forest gaps on the background as well as the fire-perturbed atmosphere. Acknowledging
the broad range in timescales relevant to fire-atmosphere interactions, analysis of both mean and

instantaneous variables was performed. A summary of the model results is presented in Fig. 12.

In the absence of fire (Fig. 12; black arrows and grayscale shading), forest gaps were found to
have a pronounced effect on mean wind and turbulence. Absent any gaps (Fig. 12a), the model ex-
hibited weak wind speeds near the surface, strong vertical wind shear across the canopy-atmosphere
interface, and two TKE maxima, a primary maximum above the canopy and a secondary maximum

near the surface. Implementation of the gap (Figs. 12b-d) yielded a recirculation zone within the
clearing, and a more homogeneous TKE field than in the surrounding forest. Downstream of the
gap, mean wind speeds were stronger than flow through a homogeneous canopy, whereas the TKE
field was similar in magnitude to the TKE field inside a homogeneous canopy.

In all cases, implementation of the fire (Fig. 12; colored arrows and shading) induced a positive

(negative) vertical velocity anomaly downstream (upstream) of the fire, with a positive horizontal
wind anomaly located immediately above the heat source (i.e., westerly inflow to the fire-induced
updraft). Whereas the positive vertical velocity anomaly showed little sensitivity to gap position,
the negative vertical velocity anomaly and horizontal wind anomaly were found to be strongest in
case CG, and weakest in case UG. The differences in horizontal and vertical wind anomaly strength

between cases were attributed to differences in background flow between the gap and the surrounding
forest. Consistent with the mean wind anomalies, TKE and turbulent mixing anomalies were largest
in case CG and weakest in case UG. Differences in TKE and turbulent mixing anomalies between
cases resulted not from differences in background flow but from differences in fire-induced buoyancy
between the gap and the surrounding forest.

This study has provided insight into the sensitivity of fire-atmosphere interactions to the presence
of gaps in forest cover. The results presented herein suggest that in order to understand how the
fire will alter wind and turbulence in a heterogeneous forest, one needs to first understand how the
forest heterogeneity itself influences the wind and turbulence fields *sans fire*. Furthermore, caution
is recommended when applying studies of fire-atmosphere interactions in homogeneous canopies





to forests with gaps. Despite the progress documented herein, much work remains. Future efforts planned includes implementing a moving fire through a forest gap, and exploring the sensitivity of fire-atmosphere interactions to canopy profile shape.

**Data availability**

The data for this paper are stored at the NCAR-Wyoming Supercomputing Center (NWSC). For data
requests, please contact Dr. Michael Kiefer at mtkiefer@msu.edu.

*Acknowledgements.* Support for this research was provided by the USDA Forest Service via Research Joint Venture Agreements 13-JV-11242306-055 and 11-JV-11242306-058. We would like to acknowledge high-performance computing support from Yellowstone (ark:/85065/d7wd3xhc) provided by NCAR's Computational and Information Systems Laboratory, sponsored by the National Science Foundation.



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



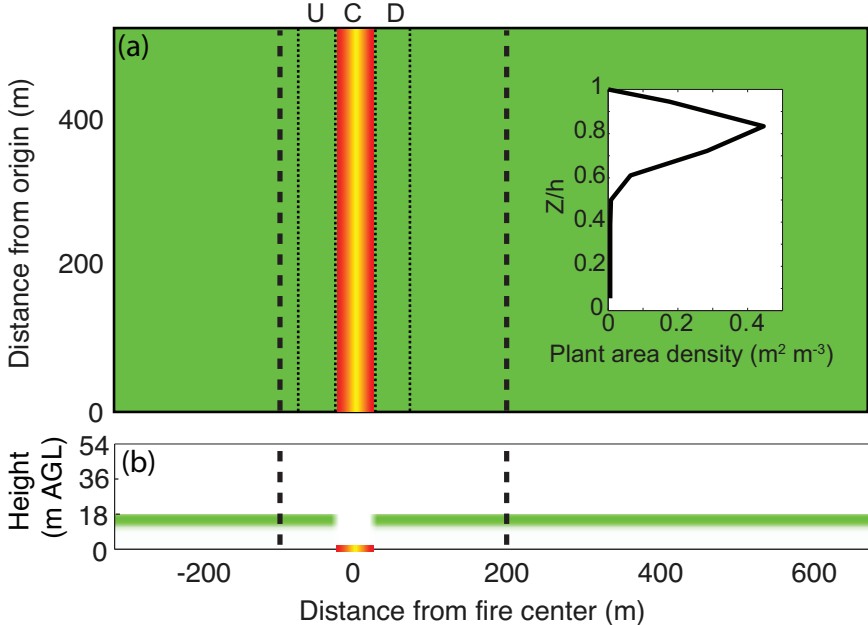

**Figure 1.** Experiment design summary, with (a) planview and (b) vertical cross-section depictions of canopy and fire. In panel (a), forest is depicted with green shading and surface heat flux in fire strip is shaded from red (weakest) → yellow (strongest). Fire strip is divided into three zones, **U**pstream, **C**enter, and **D**ownstream, with three gap cases considered: gap in the upstream zone (UG), gap in the center zone (CG), and gap in the downstream zone (DG). Inset panel in (a) depicts the plant area density profile applied at all forested points; canopy height (h) = 18 m. In panel (b), a vertical cross section of plant area density in the x-z plane is provided for case CG, along with the location of the surface fire. Thick dashed lines denote the portion of the domain displayed in subsequent figures.



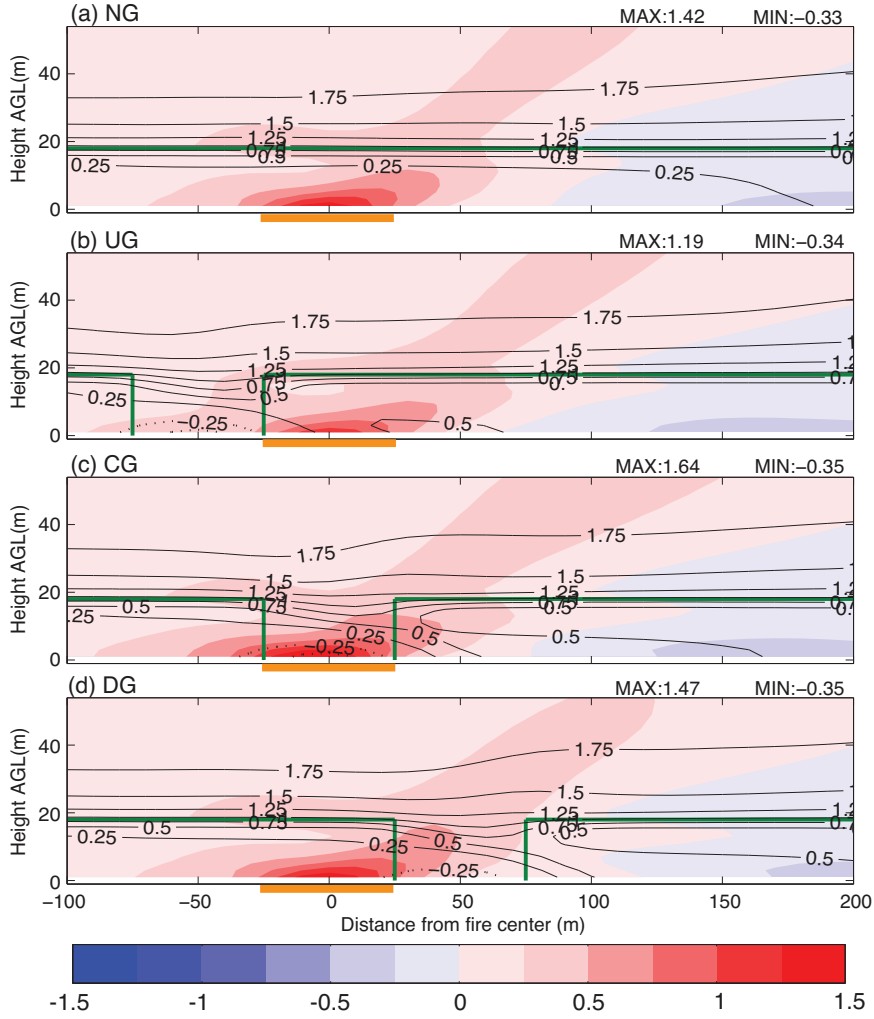

**Figure 2.** Vertical cross-sections of the u-component of the wind ($ms^{-1}$), averaged in time (30-minute) and space (along-fireline); contours denote the no-fire case (0.25 $ms^{-1}$ interval), and shading depicts the difference between the fire- and no-fire cases (0.25 $ms^{-1}$ interval). The fire zone is denoted with an orange line, and the perimeter of the forest canopy is indicated with a green line. The maximum and minimum fire–no-fire difference for each case is included above each panel. Note that only a sub-set of the domain is displayed in each panel (100 m upstream to 200 m downstream of the fire center); see Fig. 1.





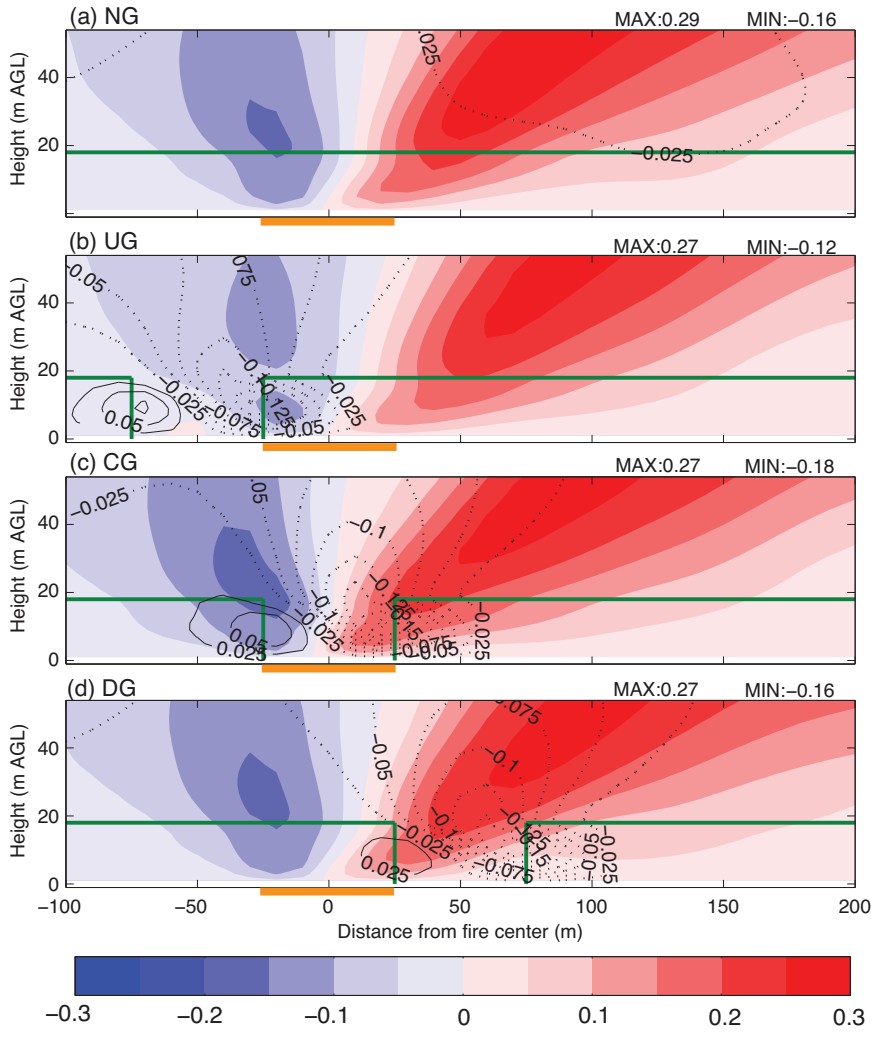

**Figure 3.** As in Fig. 2, but for vertical velocity (ms$^{-1}$); contour interval is 0.025 ms$^{-1}$ and shading interval is 0.05 ms$^{-1}$.



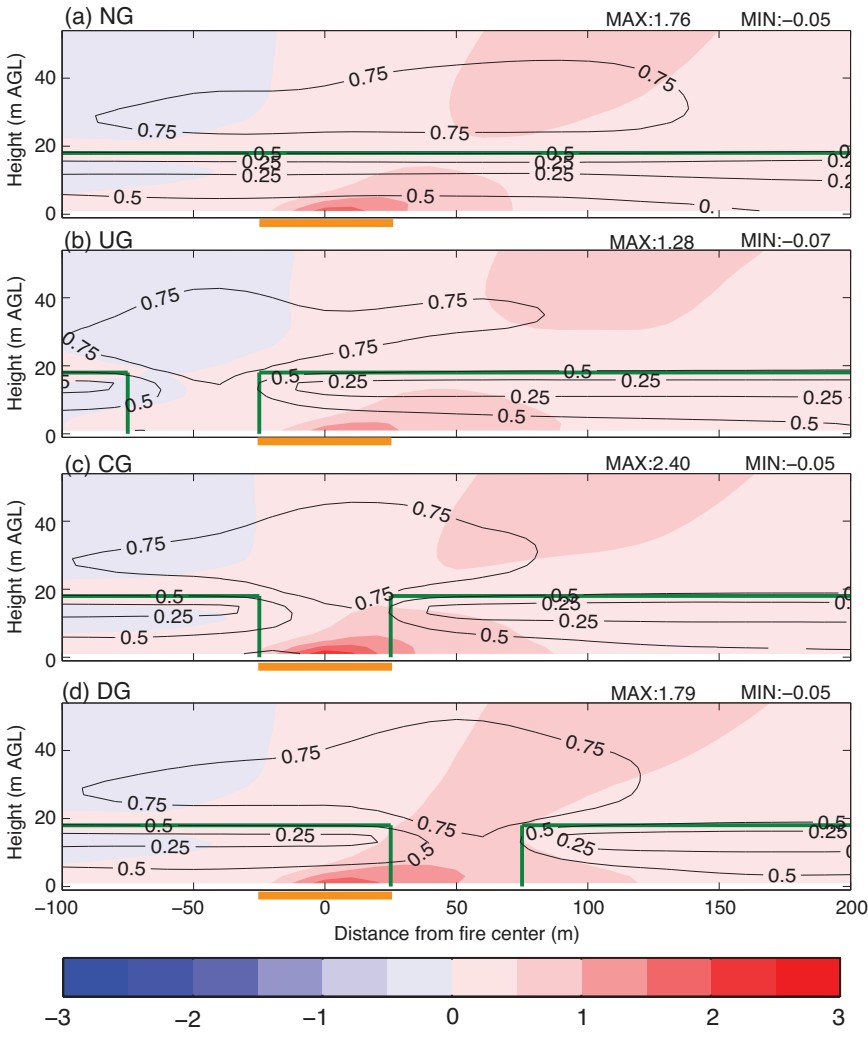

**Figure 4.** As in Fig. 2, but for TKE (m$^2$s$^{-2}$); contour interval is 0.25 m$^2$s$^{-2}$ and shading interval is 0.5 m$^2$s$^{-2}$.




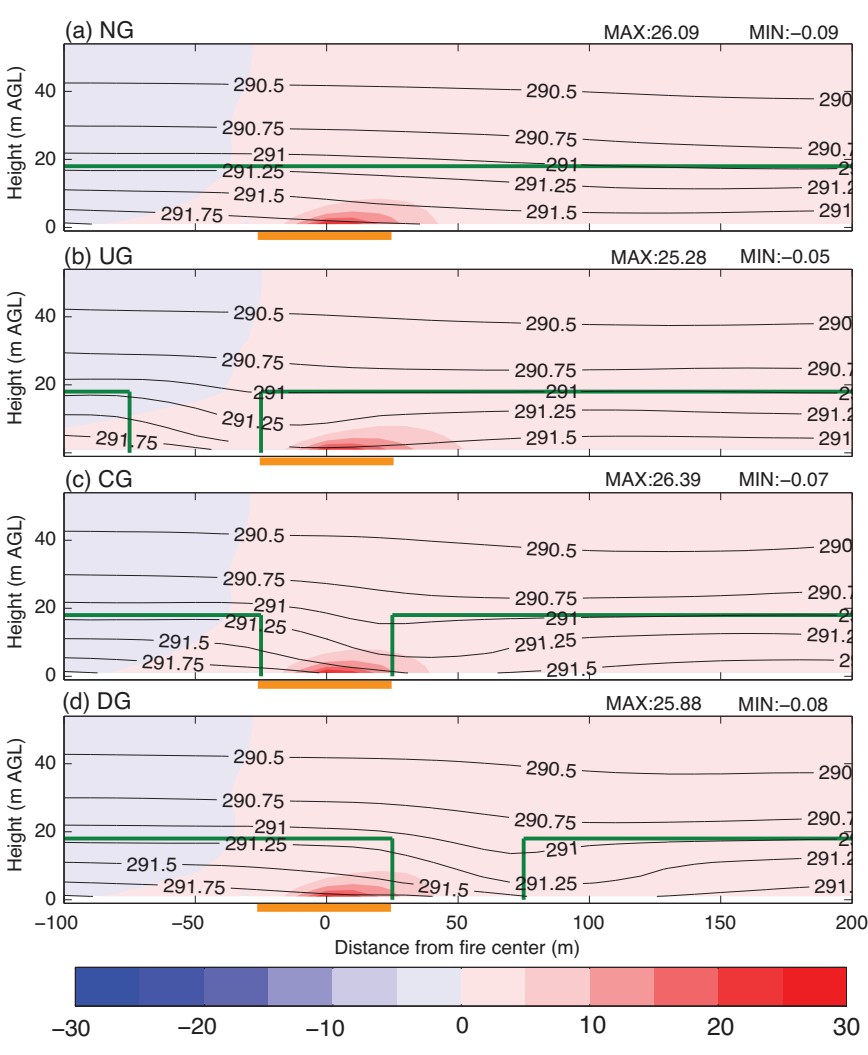

**Figure 5.** As in Fig. 2, but for temperature (K); contour interval is 0.25 K, and shading interval is 5 K.





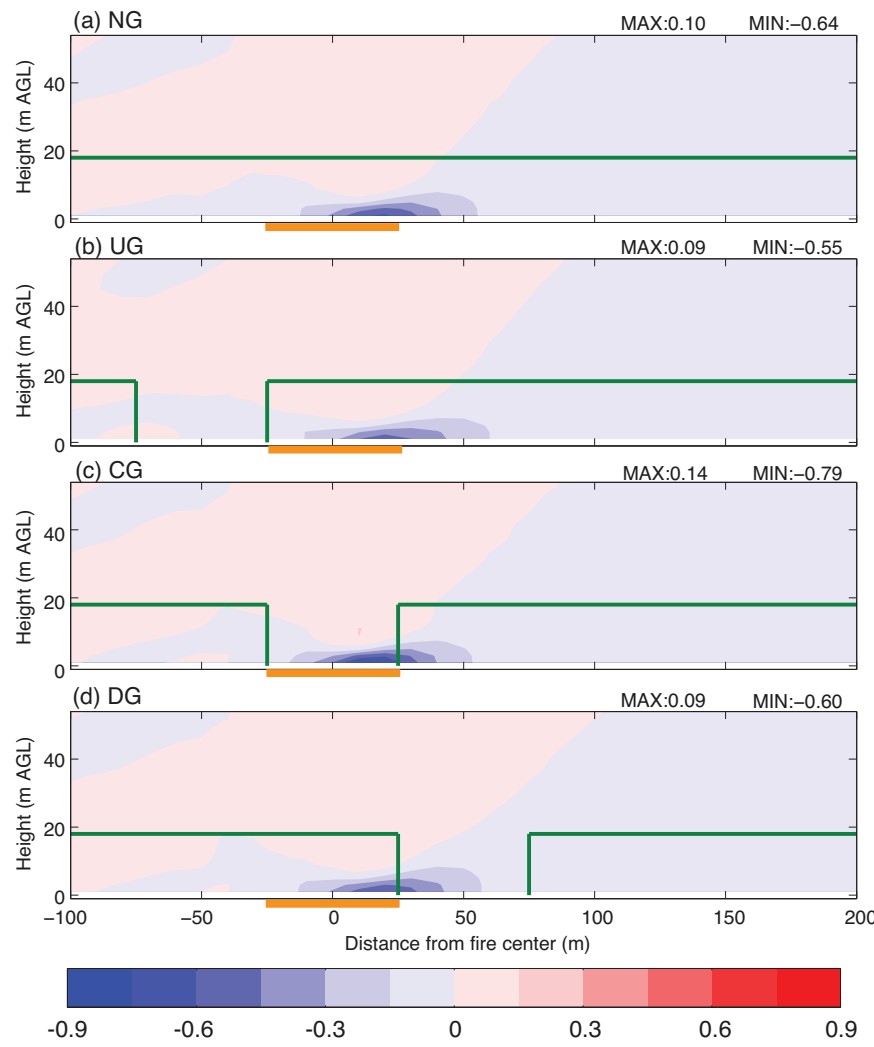

**Figure 6.** As in Fig. 2, but for the vertical gradient of the vertical turbulent heat flux ($\mathrm{Ks^{-1}}$); shading interval: $0.15\ \mathrm{Ks^{-1}}$. The no-fire case is omitted due to small values relative to fire case.





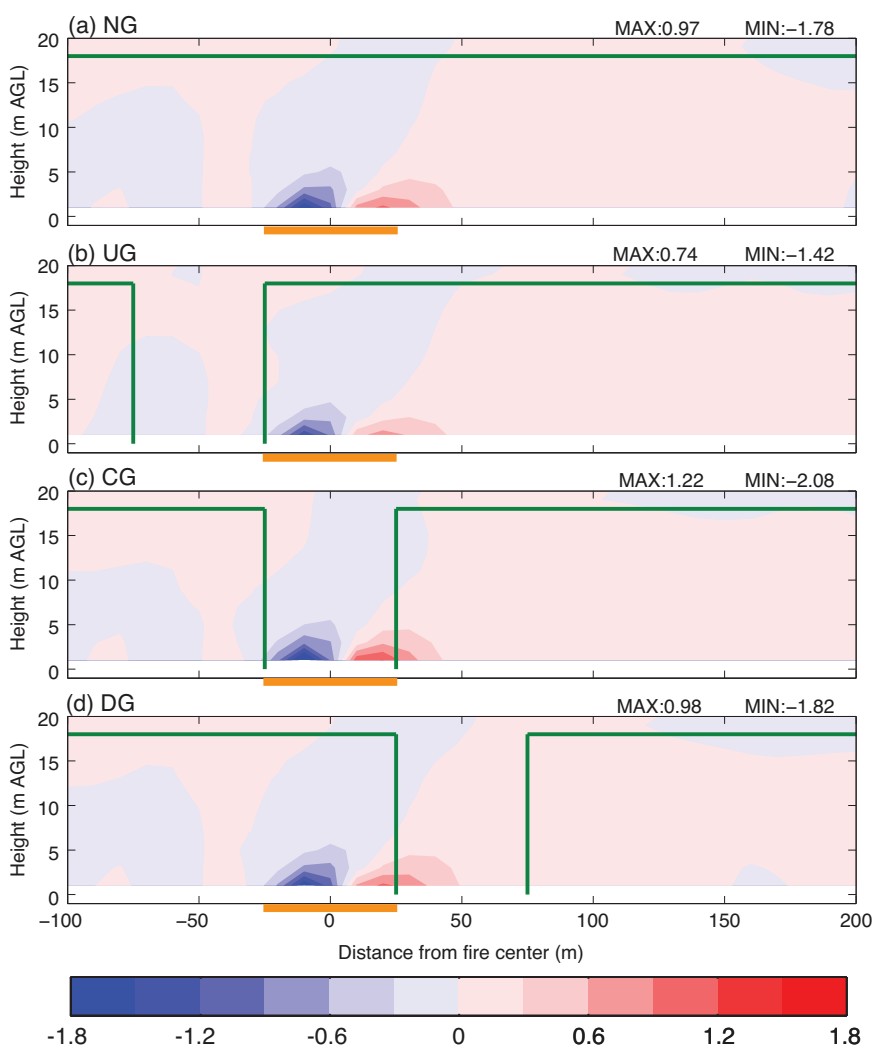

**Figure 7.** As in Fig. 2, but for the horizontal gradient of the horizontal turbulent heat flux ($\mathrm{K\,s^{-1}}$); shading interval: $0.3\ \mathrm{K\,s^{-1}}$. Note that the y-axis only extends up to 20 m AGL.





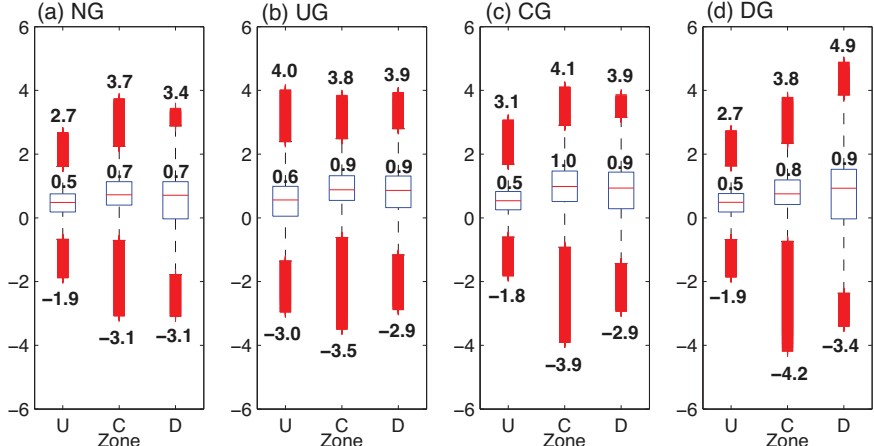

**Figure 8.** Box and whisker plots of instantaneous u-component of the wind $(\text{ms}^{-1})$ within the canopy for case (a) NG, (b) UG, (c) CG, and (d) DG. From left to right in each panel, plots correspond to grid points in the upstream, center and downstream zones (see Fig. 1 and canopy outlines in Fig. 2b-d for illustration of zones). For each box and whisker plot, the top and bottom of the box denotes the 75% ($q_3$) and 25% ($q_1$) percentiles, the horizontal red line denotes the median, the whiskers extend away from the box to the highest and lowest points not considered outliers, and the red bars extend outward from the whiskers to the maximum and minimum values. Note that outliers are determined using an interquartile range test: points are considered outliers if they are larger than $q_3 + w(q_3 - q_1)$ or smaller than $q_1 - w(q_3 - q_1)$, where $w$ is the maximum whisker length (herein, 1.5). The median, maximum, and minimum values are labeled for each plot.

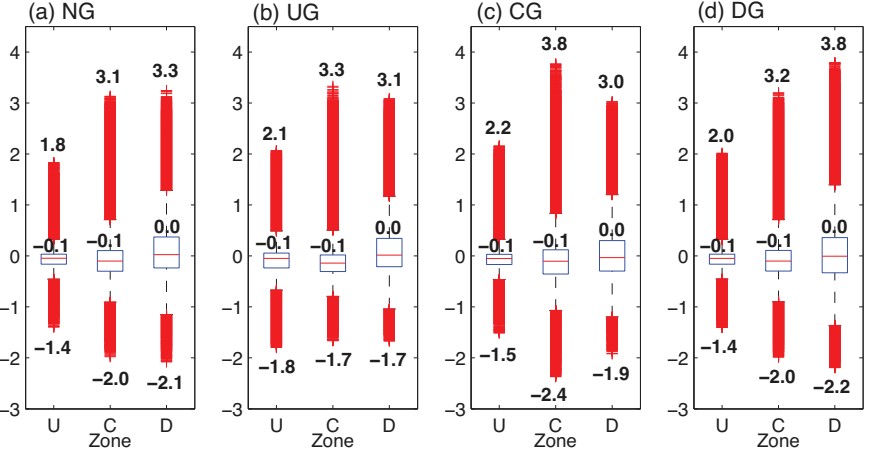

**Figure 9.** As in Fig. 8, but for vertical velocity $(\text{ms}^{-1})$.





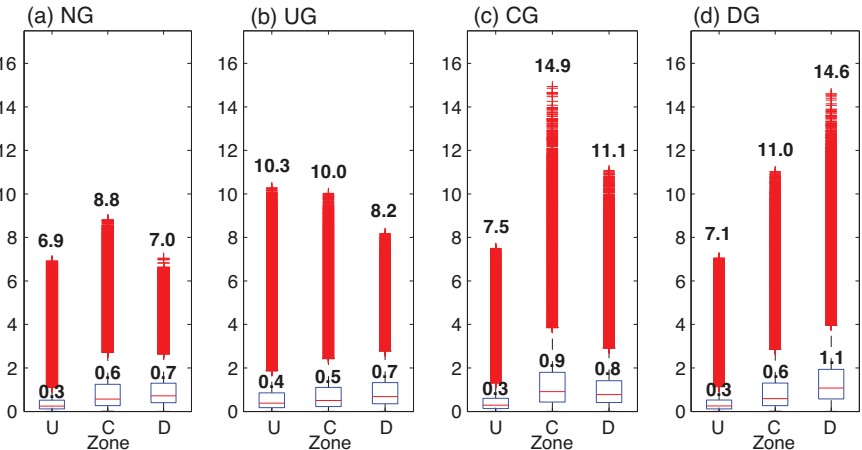

**Figure 10.** As in Fig. 8, but for TKE ($\mathrm{m^2 s^{-2}}$).

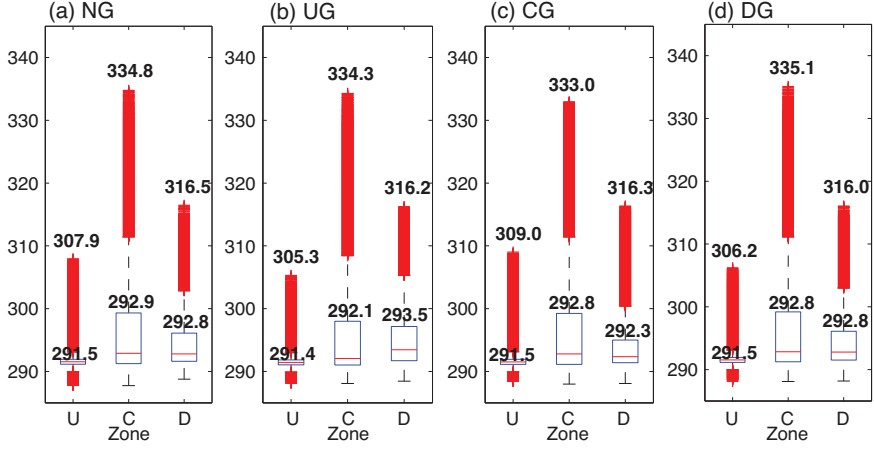

**Figure 11.** As in Fig. 8, but for temperature (K).





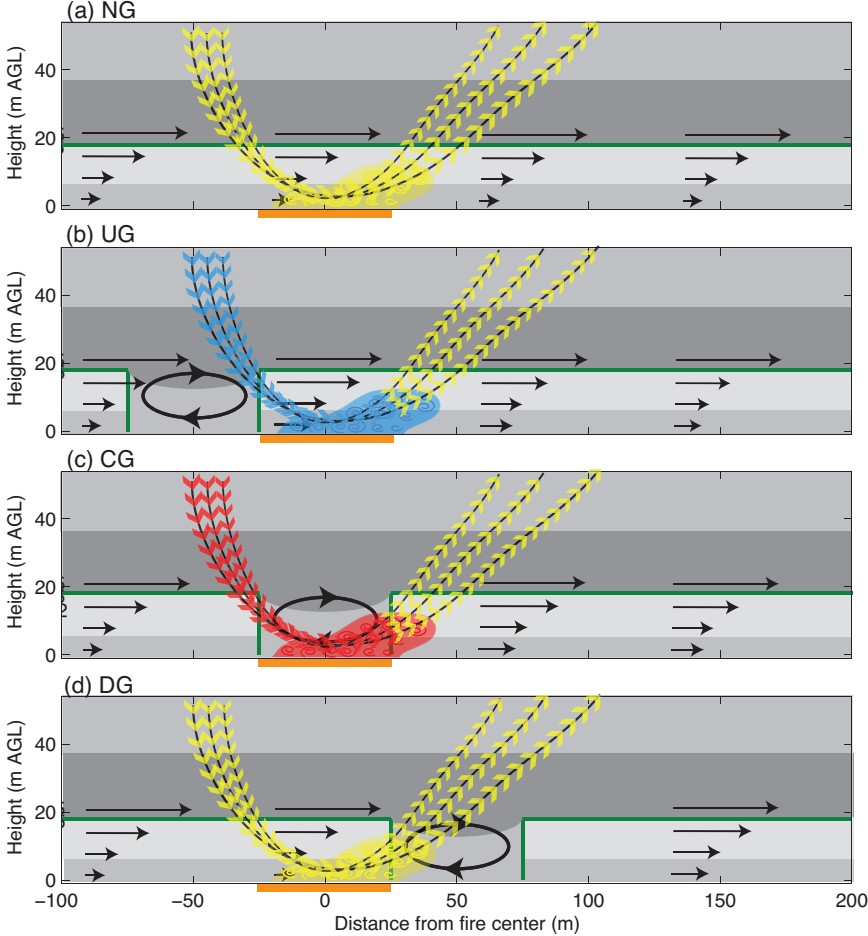

**Figure 12.** Conceptual model of fire-atmosphere interactions with different forest gap configurations. Background (i.e., no fire) state indicated with black arrows and grayscale shading: black horizontal arrows indicate the background mean u-component of the wind, black oval inside gap represents the gap recirculation zone, and shading indicates background mean TKE [light (weakest) → dark (strongest)]. Fire anomaly fields are depicted with colored arrows and shading: selected fireline-normal streamlines are indicated with colored arrows, and region of enhanced TKE and turbulent mixing of heat is represented by semi-transparent shading and embedded spirals. The magnitude of the fire anomaly is indicated by the color [blue (weakest) → yellow → red (strongest)]. As in Figs. 2-7, the fire zone is denoted with an orange line, and the perimeter of the forest canopy is indicated with a green line.