# Peer review of "A study of the influence of forest gaps on fire-atmosphere interactions"

_Atmospheric Chemistry and Physics, 2015_

## Referee Comment (RC1) · Anonymous Referee #2 · 10 Feb 2016

General comments:

This study examined the impact of two disturbance sources including forest gaps and fire sensible heat on ambient atmosphere at small scales using a series of idealized numerical experiments. Technically, it's a well-organized paper with professional analyzing and writing skills. The structure is very clear with few typo or redundancy. However, the major issue with this manuscript is about its experiment design. In another words, the numerical experiments conducted here have limited capability to address the proposed scientific question that how could forest gaps influence the interactions between fire and atmosphere. The major difference between this study and previous ones (Linn et al., 2005, 2013; Pimont et al., 2009, 2011; Potter 2012) is that there is no feedback in fire behavior to induced atmosphere disturbance, not to mention that fire in this study is highly simplified as a line of enhanced surface sensible heat flux without any other

burning processes. In this way the modeling results cannot be used directly to investigate the role of forest gaps in fire propagation, which is no doubt a very interesting topic with high scientific merits in forest management and fire protection. Under the current framework, we can only draw information about the interactive impacts from two independent disturbance sources as forest gaps and fire sensible heat on the near-surface meteorological properties. If this is what the authors intend to say (and they did make a clarification in the introduction that their focus is "on fire-atmosphere interactions in general"), then the current title of the manuscript is somewhat misleading.

Another issue with the current modeling experiments is that the differences among each sensitivity simulations are too subtle to draw definitive conclusions. Such limited sensitivity could also be a consequence of low-intensity fire perturbations in experiment design. It's suggested that an ensemble modeling experiment with high-intensity perturbations and statistical significance tests would be more convincing to evaluate the sensitivity of atmospheric responses to the two disturbances.

Overall, I recommend major revisions with additional ensemble modeling experiments before a potential publication on the ACP journal. More detailed comments are specified as follows.

Specific comments:

(1) Model description:

In section 2.1, the authors introduce multiple modifications in the model to suit the numerical experiments. These modifications include adding the drag force, the enhancement of turbulence dissipation, and radiation interception due to the canopy, etc. But no information is provided about changes in surface latent heat flux in the heterogeneous canopy due to evapotranspiration variability. In line 117, a constant albedo for forested areas is utilized. How about the albedo in these gap areas without the canopy coverage?

(2) Experiment design:

In line 158, a low-intensity fire strip is represented as a 25 kWmˆ-2 surface static sensible heat flux within a 50-m wide north-south strip. How about the length and duration of this fire strip? Previous studies (Scott and Reinhardt, 2001; Rogers, et al., 2015) have identified the distinct characteristics of low-intensity surface fire and high-intensity crown fire as well as their different impacts on the ecosystem and climate. Though the scale is different in this study, it would be still interesting to see differences of the two types of fire from a small-scale perspective.

(3) Simulation results:

In section 3.2, examination of different mean variables ends up with the same conclusion that the strongest fire anomaly occurs in the CG case and the weakest occurs in the UG case. However, this conclusion is based on the static fire assumption that fire itself has no response to ambient atmosphere disturbance. Would this conclusion be robust if we consider a more realistic fire with dynamic propagation? In the real world, fire reacts to local meteorological disturbances as well as changed fuel supplies in forest gaps with variable burning intensity, which may further interacts with ambient atmosphere in different ways like what we see here. In Pimont et al. (2011), their Fig. 5/7 demonstrates the variation of fire intensity along different forest zones. Such variability in fire intensity reaches the maximum in the Het0 case in their Fig. 7 that is similar with the forest gap setting in this study. It's suggested to consider and discuss the potential limitations related to the idealized assumption before drawing the conclusion in the context.

Technical corrections:

(1) In line 197, it should refer to "Section 3.1" instead of "Section 2.3" for a description of the averaging procedure.

References:

Linn, R. R., Winterkamp, J., Colman, J. J., Edminster, C., and Bailey, J.: Modeling Interactions between Fires and Atmosphere in Discrete Element Fuel Beds, Int. J. Wildland Fire, 14, 37–48, 2005.

Linn, R. R., Sieg, C. H., Hoffman, C. M., Winterkamp, J. L., McMillin J. D.: Modeling wind fields and fire propagation following bark beetle outbreaks in spatially-heterogeneous pinyon-juniper woodland fuel complexes, Agricultural and Forest Meteorology, 173, 139-153, 2013.

Pimont, F., Dupuy, J. L., Linn, R. R., and Dupont, S.: Validation of FIRETEC Wind-Flows over a Canopy and a Fuel-Break, Int. J. Wildland Fire, 18, 775–790, 2009.

Pimont, F., Dupuy, J. L., Linn, R. R., and Dupont, S.: Impacts of Tree Canopy Structure on Wind Flows and Fire Propagation Simulated with FIRETEC, Ann. For. Sci., 68, 523–530, 2011.

Potter, B. E.: Atmospheric interactions with wildland fire behaviour – I. Basic surface interactions, vertical profiles and synoptic structures, Int. J. Wildland Fire, 21, 779–801, 2012

Scott, J. H., Reinhardt E. D.: Assessing crown fire potential by linking models of surface and crown fire behavior, Res. Pap. RMRS-RP-29. Fort Collins, CO: U.S. Department of Agriculture, Forest Service, Rocky Mountain Research Station. 59, 2001

Rogers, B. M., Soja A. J., Goulden M. L., and Randerson J. T.: Influence of tree species on continental differences in boreal fires and climate feedbacks, Nature Geos., 8, 228-234, 2015

---

## Referee Comment (RC2) · D. Seto (Referee) · 16 Feb 2016

General comments: This research investigates the possible role forest gaps have on fire-perturbed atmospheric variables such as mean and instantaneous wind velocity, turbulence kinetic energy, and temperature using an ARPS-CANOPY model. A series of idealized simulations were conducted to examine the sensitivity of the variables modified by the low-intensity surface heat source (25 kW m-2) to the presence of forest gaps at different locations relative to the fire. Overall, I think the paper is well-written, concise, and easy to follow. The research topic will improve our current understanding in fire-atmosphere interactions in forested environments. Even though the ARPS-CANOPY model does not have a fire spread model, in my opinion it has its own merit of eliminating uncertainty and complexity embedded in current fire spread models when studying the interactions between low-intensity fire (very slow moving), atmosphere,

and forest canopy. However, I feel that more physical interpretations for what is simulated including general analysis of fire-atmosphere circulation in the forest (and forest gap) are necessary. Idealized simulations like these allow for determining the dominant forcing and feedback mechanism leading to the simulated results (and observed phenomena if any). For example, I would feel necessary that the authors address whether the fire enhances or suppresses existing recirculation in the gap; and whether updrafts and downdrafts are affected by the presence of canopy layer and strong shear layer in the gap. I think these are some of the important aspects of fire-atmosphere interaction in this study. I will present my specific comments/suggestions below, which could be worth considering before final manuscript acceptance. I recommend publication with changes made accordingly if the authors agree with my suggestions. Please feel free to reply me if my comments are unclear. Specific comments: line 212: It seems to me that the canopy layer plays a role in absorbing downward motion as shown in Figs 3a and 3d, which may contribute to lower mean Ufire-no-fire maxima (Fig.2) relative to the CG case. Is this because fire-atmosphere circulation (please see Fig. 2 in Potter 2002) is interrupted by the presence of forest canopy? I ask this question because it is related to your discussion of horizontal and vertical heat transport (line 250). Another possible explanation I can think of is the role of the strong vertical wind shear in the gap on the downward motion in case of UG (See Figs. 2b and 3b). Would it be possible to address these questions based on your simulations? I believe that such discussion would be important as the authors' main focus is on fire-atmosphere interaction in general in forested environments with gaps.

line 223: the very small differences in the positive fire anomaly found in this study may be associated with spatial averaging in y-direction. I assume that there are both positive and negative vertical velocity variations within canopy sub-layer along the lateral direction above the heat source as I imagine from Fig. 6a in Kiefer et al. (2015). If this is true, then the mean vertical velocity plots in Fig. 3 are somewhat misrepresentative of mean updraft/downdraft intensities. Even though overall structure of fire-atmosphere circulation may be true, the authors may need to inspect lateral patterns of the vertical

velocity field.

line 263 (section 3.3): It may be appropriate to present a plot showing values for the no-fire case next to the fire values (green bar/box for example). If those maximum positive and negative instantaneous values with similar magnitude were present without the fire, then the statement 'the fire has a pronounced effect on the magnitude of horizontal wind gusts... ' (line 270-271) does not hold true. I think the authors should be cautious when analyzing instantaneous max/min values although I suspect the instantaneous no-fire max/min values exceed those with the fire.

Figure 12: The conceptual model is very useful in summarizing the paper; however, the figure may be misleading because: (1) the authors did not investigate the interactions between the fire and the flow recirculation, which is one of the important features present in the forest gap. In fact, analysis of their interactions may deserve its own small section before constructing the conceptual model. Does the recirculation contribute to the overall increases in the gust and TKE? Or does the fire-induced flow circulation overcome the flow circulation in the gap? why Fig. 3b and 12b, UG case, has weakest downdrafts despite the fact that gap-induced downdraft zone meets fire-induced downdraft zone? (2) it is not clear if the model is based on mean or instantaneous variables found from your simulations? Based on Figure 10, I would think that the TKE in zone D in figur12d should be marked with red because the zone has very similar instantaneous max TKE value as the zone C in the CG case. But if I looked at Fig. 4, max TKE occurs in the CG case. Minor: line 46: are there any similarities found in this study and Pimont et al. (2009, 2011) in terms of simulated fire-atmosphere interactions inside forest gaps? If so, it would be nice to mention in your result section?

Figures 8-11: it would make the readers easier to interpret the results if you could indicate where the fire is by highlighting the zone C by orange color, just to remind them the gap location shifts relative to fire location at C.

References: Potter, B. E., 2002: A dynamics based view of atmosphere-fire interac-
tions. International Journal of Wildland Fire, 11, 247-255.

---

## Referee Comment (RC3) · Anonymous Referee #3 · 18 Feb 2016

General Comments:

This paper follows on from several recent publications that have focused on the development and use of the ARPS-Canopy model developed by Kiefer and colleagues. This new canopy model is a useful recent development, and is now being used to study the role of forest canopies in fire-atmosphere interactions. Fire-atmosphere interactions can significantly impact on wildland fire behaviour and therefore directly affect firefighter and civilian safety, so are of considerable importance.

This paper focuses on a specific application of ARPS-Canopy to look at the role of forest canopy gaps on fire to atmosphere interactions (note: no atmosphere to fire feedback is modelled). While this is an important topic that has received little prior consideration, I found the sensitivity analysis to be a little underwhelming. Of the sim-

ulations presented, consideration is primarily given to varying the forest canopy gap location relative to a time-invariant idealised representation of a low intensity fire. The sensitivity analysis reveals what in my opinion is fairly limited difference in the fire to atmosphere interaction between the four cases that include an idealised fire. As a result of this limited sensitivity, I also believe the authors need to be cautious in their conclusions. For example, the authors state, based on their results, that there is potential for forest canopy gaps to substantially affect the vertical and horizontal transport of heat away from the fire. I'm not certain that such a definitive conclusion can be drawn from this set of simulations. I am interested to know if the authors have results for a higher intensity fire, as these may show more pronounced sensitivity.

The methodology seems to be generally robust for a sensitivity analysis of this kind. The various aspects of numerical model configuration are broadly in line with what I would expect. There is good spatial resolution in the forest canopy and the model top, while fairly low, seems reasonable. The one question I would ask the authors is to clarify how the inner and outer model domains interact at the boundary, and if there are any issues regarding the transfer of momentum and turbulent kinetic energy at these boundaries. It may be useful to include additional references, if any exist, to other applications of ARPS that have used a similar high resolution configuration. While the experiment design and analysis methodology seem appropriate, they are somewhat limited in scope. The authors only consider the location of the forest canopy relative to the fire, and otherwise have an identical fire intensity and size.

The paper is well written throughout and well structured. As this is a sensitivity analysis, many of the figures directly compare the same variables between simulations. However, due to the colours chosen for the contours and limited difference in variables between simulations, in a number of figures (e.g. Figs 2-7) it is difficult to clearly discern the key differences between simulations. Additionally, the colour bars shown in Figs 2-7 appear to me to be continuous, rather than discrete, which makes it difficult to properly determine the values in those plots. Figure captions are at times overly

verbose and it could be useful to shorten where possible. In Figs 8-11, some of the labels are difficult to read due to their close proximity to the box and whisker plots. The labels are also given to two decimal points, which seems excessive.

Specific Comments:

Abstract:

1. "This study examines the impact of forest gaps on fire-atmosphere interactions" - it may be more appropriate to state that you consider only fire to atmosphere interactions, as ARPS-CANOPY does not consider the atmosphere to fire feedback.

2. The final sentence of the abstract is a little off-topic relative to what is actually discussed in the paper.

Introduction:

3. Line 81-85: Again it may be clearer to use the term "fire to atmosphere interactions" rather than "fire-atmosphere interactions". The hyphen suggests to me that there is feedback in both directions.

ARPS-CANOPY Overview:

4. Line 108-112: In these idealised simulations, does the day/night cycle affect the ground radiation budget as discussed here? I notice that the simulation is started at noon local time, so I wonder what effect this has on the simulations as they progress throughout the afternoon.

Model Configuration and Parameterization:

5. Line 134-135: If the Coriolis force is computed as a function of central latitude only, then what value for the central latitude is used? I'm assuming 40N based on details provided later. Is this term important for such a small model domain?

6. Line 143: Is there another term to describe a "rigid lid" upper boundary i.e. what kind

of boundary condition is this? I'm not very familiar with the terminology of boundary conditions, so this comment may be ignored if appropriate.

7. Line 149-150: Can you briefly quantify or more clearly describe the stable stratification above 1 km?

8. Line 153-155: How do the authors determine that a quasi-homogeneous and quasi-stationary PBL has developed? Is there some numerical test based on TKE or some other variable?

Experiment Design:

9. Line 158: "Following a 30-minute spin-up period" - what is being spun-up for 30 min? Based on the previous paragraph, is this the spin-up period in the inner domain simulation once it is initialised (i.e. after 3 hours of outer domain simulation)?

Analysis Methodology:

10. Line 181-182: Do you mention at any point the frequency of the instantaneous wind components and temperature values e.g. is it every second or minute?

Mean Variable Analysis:

11. Figs 2-3: It would be useful to see wind vectors in the xz plane showing the u and w compoent wind vectors for the "with fire" simulations, to better show that clockwise circulation within the forest canopy. It is possible to discern it from Figures 2 and 3, but not as easily.

12. Line 220: "unstable boundary layer" I thought that a neutral static stability was used for the background? Or is there some local instability induced by the idealized fire?

13. Line 227 and previously: Acronym "SGS" is only used twice after it is first defined in paper. It may be simpler just to write subgrid scale each time, as its not a particularly intuitive acronym for some readers?

14. Line 239-240: I find it interesting that a superadiabatic lapse rate is evident given the background atmospheric conditions described in the methodology. It is also not specified in which direction the weak horizontal gradient in temperature goes. I think that some additional details could be useful here.

15. Line 244: "a 4% difference" - It might be clearer to state temperature difference as an absolute value rather than as a percentage change. I personally don't have a good sense of what this 4% difference means.

16. Line 261: "potential to play a substantial role" - this statement seems too confident given the fairly limited difference between the gap and no gap cases.

Instantaneous Variable Analysis

17. Line 270: "the median is about 40% larger than in zone U" Again I would prefer to see absolute values rather than a percentage change. Or why not simply state the new median?

18. Figures 8-11: It is not clear to me precisely why the outliers are calculated for these box and whisker plots with a maximum whisker length, w, of 1.5. I also think that the description of how the outliers are determined would be better placed in the main text, such as the methodology or results section, rather than at the bottom of an already verbose figure caption.

19. Line 283: There is no quantification of distribution width or skewness, so it depends only upon a visual inspection of the box and whisker plots. I think these statements would be better with some quantified data (e.g. standard deviation or skewness, both are easily calculated) to support them.

20. Line 285-286: "the effect of the gap on the median vertical velocity value is ambiguous" I wouldn't say it is ambiguous, as you then describe the gap effects. Perhaps "inconsistent" would be a better word here?

21. Line 292-293: Conceptually, it's not particularly clear to me why a reduced-drag region (i.e. forest canopy gap) would be considerably more turbulent than the surrounding forest. I would have expected the opposite e.g. winds high above the boundary layer tend to be more laminar due to the reduced effects of surface friction. Is it related to the concepts described in the mean analysis e.g. clockwise circulation developing within the canopy gap?

Summary and Conclusions

22. Line 316: It might be useful earlier in the paper to compare 25 kW m^-2 to a typical grass or forest fire, to give some physical sense of how intense the heating is.

---

## Author Comment (AC1) · 21 Apr 2016

All responses are in red.

General comments:

This study examined the impact of two disturbance sources including forest gaps and fire sensible heat on ambient atmosphere at small scales using a series of idealized numerical experiments. Technically, it's a well-organized paper with professional analyzing and writing skills. The structure is very clear with few typo or redundancy. However, the major issue with this manuscript is about its experiment design. In another words, the numerical experiments conducted here have limited capability to address the proposed scientific question that how could forest gaps influence the interactions between fire and atmosphere. The major difference between this study and previous ones (Linn et al., 2005, 2013; Pimont et al., 2009, 2011; Potter 2012) is that there is no feedback in fire behavior to induced atmosphere disturbance, not to mention that fire in this study is highly simplified as a line of enhanced surface sensible heat flux without any other burning processes. In this way the modeling results cannot be used directly to investigate the role of forest gaps in fire propagation, which is no doubt a very interesting topic with high scientific merits in forest management and fire protection. Under the current framework, we can only draw information about the interactive impacts from two independent disturbance sources as forest gaps and fire sensible heat on the near-surface meteorological properties. If this is what the authors intend to say (and they did make a clarification in the introduction that their focus is "on fire-atmosphere interactions in general"), then the current title of the manuscript is somewhat misleading.

First, we wish to make it clear that we do not claim in the paper that the results of this study can be applied directly to the study of fire propagation.  In fact, on line 84-85, we state that our focus is "…not the specific impacts of gaps on fire behavior".  We have edited two of the sentences in the penultimate paragraph of the *Introduction* section as follows:

OLD:
"However, ARPS-CANOPY is suited to the specific goal of this study: examining how gaps in forest canopies impact fire-atmosphere interactions. Lastly, note that unlike Pimont et al. (2009) and Pimont et al. (2011), our focus is on fire-atmosphere interactions in general, and not the specific impacts of gaps on fire behavior.  Furthermore, this study considers not only how fire-atmosphere interaction in forest gaps differs from the more studied homogeneous forest case, but also whether the ability of the fire to perturb the atmosphere is sensitive to the position of the gaps relative to the fire (e.g., upstream vs. downstream gap)."

NEW:
"However, ARPS-CANOPY is suited to the specific goal of this study: examining the influence of gaps in forest canopies on atmospheric perturbations induced by a low-intensity fire.  Lastly, note that unlike Pimont et al. (2009) and Pimont et al. (2011), our focus is on fire-perturbed meteorological fields and not the specific impacts of gaps on fire behavior.  Furthermore, this

study considers whether the ability of the fire to perturb the atmosphere is sensitive to the position of the gaps relative to the fire (e.g., upstream vs. downstream gap)."

To be consistent, we have also edited one sentence near the end of the *Summary and Conclusions* section as follows:

OLD:
"This study has provided insight into the sensitivity of fire-atmosphere interactions to the presence of gaps in forest cover."

NEW:
"This study has provided insight into the sensitivity of fire-induced perturbations to the presence of gaps in forest cover."

Another issue with the current modeling experiments is that the differences among each sensitivity simulations are too subtle to draw definitive conclusions. Such limited sensitivity could also be a consequence of low-intensity fire perturbations in experiment design. It's suggested that an ensemble modeling experiment with high-intensity perturbations and statistical significance tests would be more convincing to evaluate the sensitivity of atmospheric responses to the two disturbances.

First, it is worth noting that the current version of ARPS-CANOPY is a predictive tool primarily developed for low-intensity fire applications (and other non-fire applications too).  The prediction of local smoke dispersion from low-intensity fires is made challenging by the influence of a number of interrelated factors, including near-surface meteorological conditions, local topography, vegetation structure, and atmospheric turbulence within and above vegetation layers.

This study specifically focuses on the influence of gaps in forest canopies on atmospheric perturbations induced by low-intensity fires.  This is highly relevant for understanding local plume behavior and smoke dispersion during prescribed low-intensity fires used for managing surface fuels in forested environments.  We have modified the penultimate paragraph of the *Introduction* section to state that ARPS-CANOPY was primarily developed for use with low-intensity fires, wherein transport and dispersion of smoke are especially sensitive to local vegetation, and that in this study we are examining the impact of gaps in forest cover on atmospheric perturbations above *low-intensity* fires.

That said, we agree that performing a suite of experiments with both high and low intensity fires, and calculating the statistical significance of differences between experiments, is a worthwhile endeavor.  However, there are two impediments to consider.  First, the simulations are computationally expensive, as is the post-processing.  To do statistical significance testing will require many more experiments, and thus a heavy computational workload.  Second, great uncertainty exists regarding the application of a high-intensity fire in ARPS (e.g., accuracy of computations).  In short, expanding the suite of cases to include high intensity fires and to

perform significance testing is not a trivial matter.  We will address this matter in future work.
The final sentence of the paper has been modified as follows:

OLD:
"Future efforts planned includes implementing a moving fire through a forest gap, and
exploring the sensitivity of fire-atmosphere interactions to canopy profile shape."

NEW:
"Future efforts planned include implementing a moving fire through a forest gap, exploring the
sensitivity of fire-atmosphere interactions to canopy profile shape, and performing statistical
significance testing with a larger suite of experiments, including experiments with higher-
intensity fires."

We have also added a cautionary statement near the end of the *Summary and Conclusions*
section about drawing conclusions from a limited number of experiments, with limited
sensitivity.  Lastly, we removed the qualifier "substantial" when referring to the potential role
that gaps may play in heat transport away from the fire.

Overall, I recommend major revisions with additional ensemble modeling experiments
before a potential publication on the ACP journal. More detailed comments are specified
as follows.

Specific comments:

(1) Model description:

In section 2.1, the authors introduce multiple modifications in the model to suit the
numerical experiments. These modifications include adding the drag force, the enhancement
of turbulence dissipation, and radiation interception due to the canopy, etc.
But no information is provided about changes in surface latent heat flux in the heterogeneous
canopy due to evapotranspiration variability. In line 117, a constant albedo for
forested areas is utilized. How about the albedo in these gap areas without the canopy
coverage?

No explicit changes were made in the development of ARPS-CANOPY to account for the impact
of the overlying canopy on surface evapotranspiration; however, surface evapotranspiration is
impacted by the canopy indirectly, through shading of the surface and reduction of the wind
speed due to canopy drag.  For these idealized experiments, the albedo is set to 0.3 in the gaps.
A description of this modification has been added to the text.

(2) Experiment design:

In line 158, a low-intensity fire strip is represented as a 25 kWm^-2 surface static sensible
heat flux within a 50-m wide north-south strip. How about the length and duration

of this fire strip? Previous studies (Scott and Reinhardt, 2001; Rogers, et al., 2015) have identified the distinct characteristics of low-intensity surface fire and high-intensity crown fire as well as their different impacts on the ecosystem and climate. Though the scale is different in this study, it would be still interesting to see differences of the two types of fire from a small-scale perspective.

The fireline is 500 m in length (the fire is only located in the child domain, which has a Y-dimension of 500 m). The fire is applied for a 30-min period. The missing details have been added to the text.

Regarding performing additional simulations with high intensity fires, please see our response to your General Comments (comment beginning "Another issue with the current modeling experiments…").

(3) Simulation results:

In section 3.2, examination of different mean variables ends up with the same conclusion that the strongest fire anomaly occurs in the CG case and the weakest occurs in the UG case. However, this conclusion is based on the static fire assumption that fire itself has no response to ambient atmosphere disturbance. Would this conclusion be robust if we consider a more realistic fire with dynamic propagation? In the real world, fire reacts to local meteorological disturbances as well as changed fuel supplies in forest gaps with variable burning intensity, which may further interacts with ambient atmosphere in different ways like what we see here. In Pimont et al. (2011), their Fig. 5/7 demonstrates the variation of fire intensity along different forest zones. Such variability in fire intensity reaches the maximum in the Het0 case in their Fig. 7 that is similar with the forest gap setting in this study. It's suggested to consider and discuss the potential limitations related to the idealized assumption before drawing the conclusion in the context.

We understand your concern about drawing conclusions from simulations performed with a one-way coupled model. We have added a paragraph to the end of Section 3.2 to discuss the limitations (and advantages) of the idealized modeling framework, to contrast the one-way coupled ARPS-CANOPY model with two-way coupled models such as FIRETEC and WRF-FIRE, and to caution the reader to keep the limitations (and advantages) 
[revised manuscript text omitted]

---

## Author Comment (AC2) · 21 Apr 2016

All responses are in red.

General comments:

This research investigates the possible role forest gaps have on fire-perturbed atmospheric variables such as mean and instantaneous wind velocity, turbulence kinetic energy, and temperature using an ARPS-CANOPY model. A series of idealized simulations were conducted to examine the sensitivity of the variables modified by the low-intensity surface heat source (25 kW m-2) to the presence of forest gaps at different locations relative to the fire. Overall, I think the paper is well-written, concise, and easy to follow. The research topic will improve our current understanding in fire-atmosphere interactions in forested environments. Even though the ARPSCANOPY model does not have a fire spread model, in my opinion it has its own merit of eliminating uncertainty and complexity embedded in current fire spread models when studying the interactions between low-intensity fire (very slow moving), atmosphere, and forest canopy. However, I feel that more physical interpretations for what is simulated including general analysis of fire-atmosphere circulation in the forest (and forest gap) are necessary. Idealized simulations like these allow for determining the dominant forcing and feedback mechanism leading to the simulated results (and observed phenomena if any). For example, I would feel necessary that the authors address whether the fire enhances or suppresses existing recirculation in the gap; and whether updrafts and downdrafts are affected by the presence of canopy layer and strong shear layer in the gap. I think these are some of the important aspects of fire-atmosphere interaction in this study. I will present my specific comments/suggestions below, which could be worth considering before final manuscript acceptance. I recommend publication with changes made accordingly if the authors agree with my suggestions. Please feel free to reply me if my comments are unclear.

Please see our responses to the individual comments below.

Specific comments:

line 212: It seems to me that the canopy layer plays a role in absorbing downward motion as shown in Figs 3a and 3d, which may contribute to lower mean Ufire-no-fire maxima (Fig.2) relative to the CG case. Is this because fire-atmosphere circulation (please see Fig. 2 in Potter 2002) is interrupted by the presence of forest canopy? I ask this question because it is related to your discussion of horizontal and vertical heat transport (line 250). Another possible explanation I can think of is the role of the strong vertical wind shear in the gap on the downward motion in case of UG (See Figs. 2b and 3b). Would it be possible to address these questions based on your simulations? I believe that such discussion would be important as the authors' main focus is on fire-atmosphere interaction in general in forested environments with gaps.

It is important to keep in mind that the figures in this paper present the fire anomaly, i.e., the fire minus no-fire difference.  Thus, what appears to be downward motion in Fig. 3 is strictly the difference between the fire and no-fire simulation mean vertical velocity.  It is difficult to assess the interaction of atmospheric flows and the forest canopy from 30-min mean plots of fire--no-fire simulation differences.  When we examined plots of wind vectors in the x-z plane (calculated from 1-min and y-axis-averaged u- and w-components of the wind; see our response to your comment on Figure 12 for more details) from the fire-simulation (i.e., **not** the difference field), we did not find evidence of downward motion being absorbed by the canopy layer.  One thing to keep in mind is that the canopy in these simulations is not particularly dense, with plant area index of 2.

It appears that the magnitude of the positive U fire anomaly (fire minus no-fire), as well as the negative W fire anomaly, has much to do with whether the counter-clockwise fire-induced circulation and clockwise gap-induced circulation completely overlap (CG) or only partially overlap (UG).  In the case of CG, the fire-induced downdraft develops where a gap-induced updraft existed before the fire, and the fire-induced surface west→east flow develops where gap-induced surface east→west flow existed before the fire.  Therefore, the fire anomaly field, i.e. the difference field, is largest in that case for both the u-component of the wind and vertical velocity.   In case UG, the fire-induced downdraft coincides with the location of the gap-induced downdraft, and the fire-induced west→east flow develops where relatively strong west→east flow existed before the fire, in a transition zone downstream of the gap.  Therefore, the fire anomaly field, i.e. the difference field, is smallest in that case for both variables.

As this is discussed in the existing text, we have elected not to add any additional discussion.

[No changes have been made to the manuscript in response to this comment.]

line 223: the very small differences in the positive fire anomaly found in this study may be associated with spatial averaging in y-direction. I assume that there are both positive and negative vertical velocity variations within canopy sub-layer along the lateral direction above the heat source as I imagine from Fig. 6a in Kiefer et al. (2015). If this is true, then the mean vertical velocity plots in Fig. 3 are somewhat misrepresentative of mean updraft/downdraft intensities. Even though overall structure of fire-atmosphere circulation may be true, the authors may need to inspect lateral patterns of the vertical velocity field.

In order to examine the effect of the spatial averaging step on vertical velocity anomaly differences, we have plotted horizontal cross-sections of vertical velocity anomaly at 31 m above ground level (approximate height of fire anomaly maxima and minima in Fig. 3) (Fig. R1). Note that the same procedure was used to generate Fig. R1 and Fig. 3, except the y-domain averaging step was omitted in Fig. R1.   Based on Fig. R1, it does not appear that the spatial averaging step is smoothing out substantial differences in positive fire anomaly between cases. Also, the time- and y-domain-averaged anomalies in Fig. 3 (-0.1 to +0.3 m/s) do not differ greatly from the time-averaged anomalies in Fig. R1 (-0.3 to +0.5 m/s).

We have added a statement to say that the limited differences in positive anomaly between cases is not the result of spatial averaging.

line 263 (section 3.3): It may be appropriate to present a plot showing values for the nofire case next to the fire values (green bar/box for example). If those maximum positive and negative instantaneous values with similar magnitude were present without the fire, then the statement 'the fire has a pronounced effect on the magnitude of horizontal wind gusts…' (line 270-271) does not hold true. I think the authors should be cautious when analyzing instantaneous max/min values although I suspect the instantaneous no-fire max/min values exceed those with the fire.

We understand your concern.  We have plotted the no-fire simulation instantaneous u-wind component for the NG (no gap) case (see green box and whisker plots in Fig. R2).  As can be seen, the fire has a notable impact on both the maximum, minimum, and median values of instantaneous u-wind.  For the other variables (vertical velocity, TKE, and temperature), the difference between the fire and no-fire box plots are even more pronounced (not shown).

[No changes have been made to the manuscript in response to this comment.]

Figure 12: The conceptual model is very useful in summarizing the paper; however, the figure may be misleading because: (1) the authors did not investigate the interactions between the fire and the flow recirculation, which is one of the important features present in the forest gap. In fact, analysis of their interactions may deserve its own small section before constructing the conceptual model. Does the recirculation contribute to the overall increases in the gust and TKE? Or does the fire-induced flow circulation overcome the flow circulation in the gap?

First, it is important to keep in mind that the figures in this paper present the fire anomaly, i.e., the fire minus no-fire difference. Thus, it is not absolutely correct to refer to vertical velocity anomaly maxima and minima in Fig. 3 as "updrafts" and "downdrafts".  Furthermore, it is difficult to analyze the interaction of the gap circulation and fire circulation by examining difference fields of variables averaged over a 30-min period.  The question we are able to address with the existing analyses is: do gaps influence the sign and magnitude of fire-induced perturbations above a fire, and if so, are those perturbations sensitive to the position of the gap relative to the fire?

To help answer your question about circulation interactions, we have generated plots of wind vectors in the x-z plane (calculated from 1-min and y-domain averaged u- and w-components of the wind) from the fire and no-fire simulations (**not** difference fields).  Looking at animations of these plots, it becomes clear that overall, the fire-induced flow dominates over the gap-induced circulation (see single frame in Fig. R3).  For most of the 30-minute period that the fire heat source is turned on, there is no evidence of the gap-induced circulation when looking at the wind vectors.  However, there are brief 1-min intervals when the gap circulation is dominant,

including the minute after the fire is switched on and sporadically throughout the 30-minute period (not shown).  It is also worth noting that even in the no-fire simulation, the gap circulation itself is not entirely steady, disappearing altogether for one or two minutes at a time.  This occurs due to the presence of transient features in the boundary layer that temporarily overwhelm the circulation anchored to the gap.  In summary, averaged over short timescales (e.g., 1-2 min), TKE and gusts are potentially influenced by the gap circulation, but averaged over longer timescales (e.g., 30 min), the influence of the gap circulation is likely minimal.

Although we are not adding any new figures to the paper, we have added a paragraph to the *Results and Discussion* section, following the mean TKE paragraph, to discuss potential interactions (rather than adding a separate sub-section as you suggested).

why Fig. 3b and 12b, UG case,
has weakest downdrafts despite the fact that gap-induced downdraft zone meets fire induced downdraft zone?

Examining the time and space-averaged (30-min, y-domain) vertical velocity in the four fire simulations (not the fire anomaly), we find that the strongest downdraft is actually found in case UG, on the eastern edge of the gap.  But this is precisely where the gap-induced downdraft was located before the fire was switched on.  Thus, the fire-induced downdraft reinforces a pre-existing gap-induced downdraft.  The fire—no-fire vertical velocity difference is actually smallest in that case, even though the strongest downdraft occurs in the UG fire simulation.

[No changes have been made to the manuscript in response to this comment.]

(2) it is not clear if the model is based on mean or instantaneous
variables found from your simulations? Based on Figure 10, I would think that
the TKE in zone D in figur12d should be marked with red because the zone has very
similar instantaneous max TKE value as the zone C in the CG case. But if I looked
at Fig. 4, max TKE occurs in the CG case.

The model is based on mean variables, not instantaneous.  This detail has been added to the text where the conceptual model is first introduced.

Minor: line 46: are there any similarities
found in this study and Pimont et al. (2009, 2011) in terms of simulated fire-atmosphere interactions inside forest gaps? If so, it would be nice to mention in your result section?

We have added a sentence to the end of the first paragraph in the *Instantaneous Variable Analysis* sub-section to comment on similarities in findings to the Pimont et al. studies.  In addition, we have added a paragraph to the end of the *Mean Variable Analysis* sub-section to contrast the idealized modeling framework in this study to coupled-fire atmosphere models like FIRETEC (used in Pimont et al. studies).  This paragraph was added in response to a comment

from another reviewer.

Figures 8-11: it would make the readers easier to interpret the results if you could indicate where the fire is by highlighting the zone C by orange color, just to remind them the gap location shifts relative to fire location at C.

An orange bar has been added to the bottom center of each panel to indicate the fire's location in zone C.  The caption for Fig. 8 has been modified accordingly.

References: Potter, B. E., 2002: A dynamics based view of atmosphere-fire interactions. International Journal of Wildland Fire, 11, 247-255.

[Figure]

Fig. R1. Horizontal cross sections of vertical velocity fire anomaly at 31 m above ground level, averaged in time (30-minute). Shading denotes the difference between the fire and no-fire cases (0.05 ms$^{-1}$ interval). Dashed box indicates the location of the gap, and the orange box denotes the location of the fire zone.

[Figure]

Fig. R2.  As in Fig. 8a, but with box and whisker plots from the no-fire simulation (green lines and bars).

[Figure]

Fig. R3. Vertical cross-sections of 1-min mean temperature (K) with wind vectors in the X-Z plane overlaid (computed from 1-min mean u- and w-wind components), 12 minutes after fire initiation. Temperatures in fire simulation are shaded, and temperatures in no-fire simulation are contoured. Note that all fields are averaged in the y-direction. Black arrows are from no-fire simulation, and red arrows are from fire-simulation. Maximum 1-min mean temperature in fire simulation indicated above each panel.

[revised manuscript text omitted]

---

## Author Comment (AC3) · 21 Apr 2016

All responses are in red.

General Comments:

This paper follows on from several recent publications that have focused on the development and use of the ARPS-Canopy model developed by Kiefer and colleagues. This new canopy model is a useful recent development, and is now being used to study the role of forest canopies in fire-atmosphere interactions. Fire-atmosphere interactions can significantly impact on wildland fire behaviour and therefore directly affect firefighter and civilian safety, so are of considerable importance.

This paper focuses on a specific application of ARPS-Canopy to look at the role of forest canopy gaps on fire to atmosphere interactions (note: no atmosphere to fire feedback is modelled). While this is an important topic that has received little prior consideration, I found the sensitivity analysis to be a little underwhelming. Of the simulations presented, consideration is primarily given to varying the forest canopy gap location relative to a time-invariant idealised representation of a low intensity fire. The sensitivity analysis reveals what in my opinion is fairly limited difference in the fire to atmosphere interaction between the four cases that include an idealised fire. As a result of this limited sensitivity, I also believe the authors need to be cautious in their conclusions. For example, the authors state, based on their results, that there is potential for forest canopy gaps to substantially affect the vertical and horizontal transport of heat away from the fire. I'm not certain that such a definitive conclusion can be drawn from this set of simulations. I am interested to know if the authors have results for a higher intensity fire, as these may show more pronounced sensitivity.

The current four-experiment study constitutes the initial phase of a larger study examining sensitivity of perturbations to canopy profile shape, fire intensity, and fire spread (e.g., backing vs. head fires). These items are included as future work at the end of the final paragraph of the manuscript (we have added fire intensity to the list). We have removed the qualifier "substantial" to avoid overstating the results (see your comment #16). Also, we have added a cautionary statement in the conclusions section about drawing conclusions from a limited number of experiments, with limited sensitivity.

The methodology seems to be generally robust for a sensitivity analysis of this kind. The various aspects of numerical model configuration are broadly in line with what I would expect. There is good spatial resolution in the forest canopy and the model top, while fairly low, seems reasonable. The one question I would ask the authors is to clarify how the inner and outer model domains interact at the boundary, and if there are any issues regarding the transfer of momentum and turbulent kinetic energy at these boundaries. It may be useful to include additional references, if any exist, to other applications of ARPS that have used a similar high resolution configuration. While the experiment design and analysis methodology seem appropriate, they are somewhat

limited in scope. The authors only consider the location of the forest canopy relative to the fire, and otherwise have an identical fire intensity and size.

For the inner model domain, we use time-dependent one-way lateral boundary conditions, that is, values of prognostic variables at the lateral boundary points of the inner domain are obtained from the outer domain simulation (updated every five minutes), but the outer domain simulation is completely independent of the inner domain simulation.  For more information, please see the ARPS User's Guide (http://arps.ou.edu/arpsdoc.html).  We have inserted "one-way lateral" where boundary conditions for the inner domain are first introduced in the text, to clarify the nature of the lateral boundary conditions. We are unaware of any issues with the transfer of momentum and TKE across lateral boundaries in ARPS.  Other studies that have used ARPS with comparable spatial resolution include Dupont and Brunet (2008) and Michioka and Chow (2008) [note that the latter also used one-way nesting].  We have added these references to the text where the grid spacing is introduced.  Regarding fire intensity experiments, see our response to your Scientific Significance comments (previous paragraph).

The paper is well written throughout and well structured. As this is a sensitivity analysis, many of the figures directly compare the same variables between simulations. However, due to the colours chosen for the contours and limited difference in variables between simulations, in a number of figures (e.g. Figs 2-7) it is difficult to clearly discern the key differences between simulations. Additionally, the colour bars shown in Figs 2-7 appear to me to be continuous, rather than discrete, which makes it difficult to properly determine the values in those plots. Figure captions are at times overly verbose and it could be useful to shorten where possible. In Figs 8-11, some of the labels are difficult to read due to their close proximity to the box and whisker plots. The labels are also given to two decimal points, which seems excessive.

Editor and reviewer note: We made changes to the manuscript in response to these comments during the access review stage:

[We experimented with different color schemes, and came to the conclusion that the blue->white->red color scheme is most ideal due to its simplicity and easy adaptability to each variable.  However, we have reduced the number of shades in each plot to 12, to help the reader distinguish between shades, and we modified the color scheme so that values one interval either side of zero are shaded (i.e., no whitespace).  We have also made the color bars discrete.  Finally, the captions in Figs. 1 and 8 have been made less verbose, the positions of the labels in Figs. 8-11 have been adjusted to improve readability, and we now display only one digit to the right of the decimal point.]

Specific Comments:

Abstract:
1. "This study examines the impact of forest gaps on fire-atmosphere interactions" - it may be more appropriate to state that you consider only fire to atmosphere interactions,

as ARPS-CANOPY does not consider the atmosphere to fire feedback.

We appreciate your concern about our use of the term "fire-atmosphere interactions" to describe this study. Rather than use alternate wording like "fire to atmosphere interactions" we prefer to tone down the use of "fire-atmosphere interactions" in the text and be more specific about what we are examining in this study (changes were made, where appropriate, in the Abstract, Introduction, and Summary and Conclusions section). We have edited the second sentence in the abstract as follows:

OLD:
"This study examines the impact of forest gaps on fire-atmosphere interactions using the ARPS-CANOPY model…"

NEW:
"This study examines the influence of gaps in forest canopies on atmospheric perturbations induced by a low-intensity fire using the ARPS-CANOPY model…"

2. The final sentence of the abstract is a little off-topic relative to what is actually discussed in the paper.

We do not feel that the sentence in question is off-topic. This is a study of how gaps influence atmospheric perturbations above fires, and gaps fall under the broader heading of "forest heterogeneity". We prefer to leave the sentence intact.

[No changes have been made to the manuscript in response to this comment.]

Introduction:
3. Line 81-85: Again it may be clearer to use the term "fire to atmosphere interactions" rather than "fire-atmosphere interactions". The hyphen suggests to me that there is feedback in both directions.

Similar to our response to comment 1, we have edited the sentences on lines 81-85 as follows:

OLD:
"However, ARPS-CANOPY is suited to the specific goal of this study: examining how gaps in forest canopies impact fire-atmosphere interactions. Lastly, note that unlike Pimont et al. (2009) and Pimont et al. (2011), our focus is on fire-atmosphere interactions in general, and not the specific impacts of gaps on fire behavior. Furthermore, this study considers not only how fire-atmosphere interaction in forest gaps differs from the more studied homogeneous forest case, but also whether the ability of the fire to perturb the atmosphere is sensitive to the position of the gaps relative to the fire (e.g., upstream vs. downstream gap)."

NEW:

"However, ARPS-CANOPY is suited to the specific goal of this study: examining the influence of gaps in forest canopies on atmospheric perturbations induced by a low-intensity fire. Lastly, note that unlike Pimont et al. (2009) and Pimont et al. (2011), our focus is on fire-perturbed meteorological fields and not the specific impacts of gaps on fire behavior. Furthermore, this study considers whether the ability of the fire to perturb the atmosphere is sensitive to the position of the gaps relative to the fire (e.g., upstream vs. downstream gap)."

ARPS-CANOPY Overview:
4. Line 108-112: In these idealised simulations, does the day/night cycle affect the ground radiation budget as discussed here? I notice that the simulation is started at noon local time, so I wonder what effect this has on the simulations as they progress throughout the afternoon.

When performing a full-physics simulation, the day/night cycle does affect the ground radiation budget. However, for these idealized simulations, steady net radiation of 520 Wm$^{-2}$ is applied at canopy top throughout the simulation. So, the diurnal cycle of net radiation has no effect on these simulations.

[No changes have been made to the manuscript in response to this comment.]

Model Configuration and Parameterization:
5. Line 134-135: If the Coriolis force is computed as a function of central latitude only, then what value for the central latitude is used? I'm assuming 40N based on details provided later. Is this term important for such a small model domain?

40$^o$ north latitude is specified (arbitrary); this detail has been added to the text. The term is likely negligible for the spatial and temporal scales considered herein, but was included nevertheless.

6. Line 143: Is there another term to describe a "rigid lid" upper boundary i.e. what kind of boundary condition is this? I'm not very familiar with the terminology of boundary conditions, so this comment may be ignored if appropriate.

The "rigid lid" upper boundary condition is equivalent to the "wall" lateral boundary condition. This is a fairly standard term (a Google search of "rigid lid upper boundary condition" returned 589 results).

[No changes have been made to the manuscript in response to this comment.]

7. Line 149-150: Can you briefly quantify or more clearly describe the stable stratification above 1 km?

The Brunt-Väisälä frequency, N, is 0.013 s$^{-1}$. This detail has been added to the manuscript.

8. Line 153-155: How do the authors determine that a quasi-homogeneous and quasistationary PBL has developed? Is there some numerical test based on TKE or some other variable?

We examined time series of mean temperature and wind speed within the outer domain, along with the standard deviation of these variables, and made a subjective assessment of quasi-stationarity (e.g., mean wind speed in the domain exhibits steady behavior after about 3 hours of model integration). Quasi-homogeneity was subjectively determined based on a review of horizontal cross-sections of mean temperature and wind speed.

[No changes have been made to the manuscript in response to this comment.]

Experiment Design:
9. Line 158: "Following a 30-minute spin-up period" - what is being spun-up for 30 min? Based on the previous paragraph, is this the spin-up period in the inner domain simulation once it is initialised (i.e. after 3 hours of outer domain simulation)?

ARPS requires that the child domain be initialized with zero subgrid-scale turbulence. The 30-min spin up period allows time for turbulence to develop. The first sentence refers to the 30-min spin-up period in the inner domain simulation (a 3-hour spin-up period is used in the outer domain simulation, as described in the last paragraph of Section 2.2).

[No changes have been made to the manuscript in response to this comment.]

Analysis Methodology:
10. Line 181-182: Do you mention at any point the frequency of the instantaneous wind components and temperature values e.g. is it every second or minute?

The instantaneous wind components are dumped out every second (i.e., 1 Hz frequency). This detail has been added to the text.

Mean Variable Analysis:
11. Figs 2-3: It would be useful to see wind vectors in the xz plane showing the u and w wind compoent wind vectors for the "with fire" simulations, to better show that clockwise circulation within the forest canopy. It is possible to discern it from Figures 2 and 3, but not as easily.

We agree that wind vectors would help the reader visualize the gap and fire-induced circulations. However, we put much effort into producing a modified Fig. 2 with the wind vectors overlaid and, unfortunately, found it impossible to prevent the figure from becoming too cluttered.

[No changes have been made to the manuscript in response to this comment.]

12. Line 220: "unstable boundary layer" I thought that a neutral static stability was used for the background? Or is there some local instability induced by the idealized fire?

A neutrally-stratified profile is used for the background state, however, net radiation of 520 W m$^{-2}$ is applied at the canopy top (or ground surface in the gaps). This results in an unstable boundary layer. Information about the radiation flux was provided in the outer domain description in Section 3.2, but was unintentionally omitted from the inner domain description. This detail has been added to the manuscript to avoid misunderstanding.

13. Line 227 and previously: Acronym "SGS" is only used twice after it is first defined in paper. It may be simpler just to write subgrid scale each time, as its not a particularly intuitive acronym for some readers?

SGS has been replaced with sub-grid scale.

14. Line 239-240: I find it interesting that a superadiabatic lapse rate is evident given the background atmospheric conditions described in the methodology. It is also not specified in which direction the weak horizontal gradient in temperature goes. I think that some additional details could be useful here.

Regarding the presence of a superadiabatic lapse rate, see our response to comment #12. The sign of the horizontal temperature gradient (negative, i.e., temperature decreases from left to right) has been added to the text.

15. Line 244: "a 4% difference" - It might be clearer to state temperature difference as an absolute value rather than as a percentage change. I personally don't have a good sense of what this 4% difference means.

The anomaly difference is 1.1 K. We have replaced "…about a 4% difference…" with "…a 1.1K (about 4%) difference…".

16. Line 261: "potential to play a substantial role" - this statement seems too confident given the fairly limited difference between the gap and no gap cases.

In response to this comment, and your Scientific Significance comments, we have removed the word "substantial" from line 161 as well as the abstract.

Instantaneous Variable Analysis

17. Line 270: "the median is about 40% larger than in zone U" Again I would prefer to see absolute values rather than a percentage change. Or why not simply state the new median?

The difference in the median is 0.2 m s$^{-1}$. The absolute value is now included in the text.

18. Figures 8-11: It is not clear to me precisely why the outliers are calculated for these box and whisker plots with a maximum whisker length, w, of 1.5. I also think that the description of how the outliers are determined would be better placed in the main text, such as the methodology or results section, rather than at the bottom of an already verbose figure caption.

We used Matlab analysis software to generate the plots in this study. Matlab has a built-in box and whisker plot function that defines outliers as values that are more than w times the interquartile range away from the 25$^{th}$ ($q_1$) and 75$^{th}$ ($q_3$) percentiles [i.e., $q_3 + w*(q_3 - q_1)$ or $q_1 - w*(q_3-q_1)$]. In Matlab, the default value for w is 1.5. The documentation for this function states that "The default of 1.5 corresponds to approximately +/−2.7$\sigma$ and 99.3 percent coverage if the data are normally distributed."

The description of the outlier methodology has been moved to the Analysis Methodology subsection, and the new detail about the choice of w (i.e., +/- 2.7 $\sigma$ coverage) has been added to the text.

19. Line 283: There is no quantification of distribution width or skewness, so it depends only upon a visual inspection of the box and whisker plots. I think these statements would be better with some quantified data (e.g. standard deviation or skewness, both are easily calculated) to support them.

We have computed skewness and added the information to the text to better support our statements.

20. Line 285-286: "the effect of the gap on the median vertical velocity value is ambiguous" I wouldn't say it is ambiguous, as you then describe the gap effects. Perhaps "inconsistent" would be a better word here?

"Ambiguous" changed to "inconsistent".

21. Line 292-293: Conceptually, it's not particularly clear to me why a reduced-drag region (i.e. forest canopy gap) would be considerably more turbulent than the surrounding forest. I would have expected the opposite e.g. winds high above the boundary layer tend to be more laminar due to the reduced effects of surface friction. Is it related to the concepts described in the mean analysis e.g. clockwise circulation developing within the canopy gap?

The drag force yields relatively weak mean wind speeds within the canopy, with a layer of pronounced vertical wind shear (and consequently, turbulence production) in the upper portion of the canopy. Deep within the canopy, any turbulence production and/or transport into that layer is countered by canopy drag. Removing the vegetation in a small area of the larger forest locally eliminates the canopy drag effect, allowing turbulence production and transport to

contribute to TKE without the canopy sink term.

[No changes have been made to the manuscript in response to this comment.]

Summary and Conclusions
22. Line 316: It might be useful earlier in the paper to compare 25 kW mˆ-2 to a typical
grass or forest fire, to give some physical sense of how intense the heating is.

We have added a sentence to compare 25 kW m$^{-2}$ to values observed during low-intensity fire
field campaigns (e.g., RxCadre). The value used herein is within the envelope of observed
values and is considered representative of a low-intensity fire.

[revised manuscript text omitted]